# Impact of melt water on high latitude early Last Interglacial climate

Emma J. Stone[1], Emilie Capron[2,3], Daniel J. Lunt[1], Antony J. Payne[1], Joy S. Singarayer[4], Paul J. Valdes[1] and Eric W.Wolff[5]

[1] BRIDGE, School of Geographical Sciences, University of Bristol, Bristol, UK
[2] British Antarctic Survey, Cambridge, UK
[3] Centre for ice and Climate, Niels Bohr Institute, University of Copenhagen, Copenhagen, Denmark
[4] Department of Meteorology, University of Reading, Reading, UK
[5] Department of Earth Sciences, University of Cambridge, Cambridge, UK

*Correspondence to*: E. J. Stone (emma.j.stone@bristol.ac.uk)

**Abstract.** Recent data compilations of the early Last Interglacial period have indicated a bipolar temperature response at 130 ka, with colder-than-present temperatures in the North Atlantic and warmer-than-present temperatures in the Southern Ocean and over Antarctica. However, climate model simulations of this period have been unable to reproduce this response, when only orbital and greenhouse gas forcings are considered in a climate model framework. Using a full complexity General Circulation Model we perform climate model simulations representative of 130 ka conditions which include a magnitude of freshwater forcing derived from data at this time. We show that this meltwater from the remnant Northern Hemisphere ice-sheets during the glacial-interglacial transition produces a modelled climate response similar to the observed colder-than-present temperatures in the North Atlantic at 130 ka, and also results in warmer-than-present temperatures in the Southern Ocean via the bipolar seesaw mechanism. Further simulations in which the West Antarctic ice sheet is also removed lead to warming in East Antarctica and the Southern Ocean but do not appreciably improve the model-data comparison. This integrated model-data approach provides evidence that Northern Hemisphere freshwater forcing is an important player in the evolution of early Last Interglacial climate.

## 1 Introduction

Understanding the climate feedback processes that occur in the high latitude regions is essential because they are particularly sensitive to changes in radiative forcing and act as amplifiers of climate change (Vaughan et al., 2013). Peak high latitude temperatures were several degrees warmer during the Last Interglacial (LIG, approximately 129-116 thousand years ago, ka, based on eustatic sea level variations, Masson-Delmotte et al. 2013) (Clark and Huybers, 2009; Masson-Delmotte et al., 2011; Otto-Bliesner et al., 2006; Sime et al., 2009; Turney and Jones, 2010) and maximum global sea level was 6 to 9 m higher than today (Dutton et al., 2015; Dutton and Lambeck, 2012; Kopp et al., 2009). Thus, the LIG represents an ideal case study to understand and test the climate mechanisms that operate under warm climates. The LIG, however, should not be considered a direct analogue for future climate due to the difference in primary forcing mechanisms of seasonal astronomical changes versus greenhouse gas (GHG) changes to explain the observed warmth.

Until recently, climate model simulations of the LIG were typically compared with a data synthesis for surface temperature consisting of one single snapshot representing the warmest temperature anomalies for the whole LIG (Lunt et al., 2013; McKay et al., 2011; Otto-Bliesner et al., 2013). In particular, the annual surface temperature data synthesis from Turney and Jones (2010) illustrates the large-scale spatial pattern in peak LIG warmth but does not provide a global temporal climatic evolution

due to the difficulty in obtaining robust and coherent LIG chronologies (Govin et al., 2015). Such a compilation of LIG maximum warmth, as in the approach of Turney and Jones, neglects any potential asynchronous temperature changes between regions while previous studies (Bauch et al., 2011; CLIMAP Project Members, 1984; Govin et al., 2012; Ruddiman et al., 1980; Van Nieuwenhove et al., 2011; Winsor et al., 2012), though limited to only a few records, have provided evidence of hemispheric surface temperature asynchrony during the early LIG.

A new LIG compilation (Capron et al., 2014) of surface temperature changes has been produced for the high latitude oceans (latitudes northward of 40°N and southward of 40°S) and polar ice-sheets. In contrast to previous LIG datasets, this new data synthesis benefits from a coherent temporal framework between marine and ice core records. It thus provides the first spatio-temporal description of the climate between 135 and 110 ka. In particular, surface temperature anomalies have been calculated for four time windows: 114-116, 119-121, 124-126 and 129-131 ka, referred to as the data-based 115, 120, 125 and 130 ka

time slices. These four time slices are associated with quantitative estimates of temperature errors, including the error in the reconstructed sea surface temperature (SST) and the propagation of dating uncertainties: the 2σ uncertainty on SST anomalies is 2.6ºC on average and 1.5ºC for Antarctic surface temperatures (see Capron et al. (2014) for methodological details and 2σ uncertainty estimates for individual records). Note that Antarctic annual surface air temperature reconstructions are estimated based on the water isotopic records after correction for sea water isotopic composition and moisture source correction using

deuterium excess data (Masson-Delmotte et al. 2011). Capron et al. (2014) consider an error of 1.5°C associated with these reconstructions. It accounts for the uncertainty associated with this method and also partially accounts for the uncertainty associated with possible impacts of changes in seasonality of precipitation on the reconstructions, which remains difficult to quantify in ice core data (Masson-Delmotte et al. 2011).

The data-based 130 ka time slice indicates robust new insights into the early LIG climate with asynchronous maximum summer

temperature changes relative to present day between the two hemispheres where the Southern Ocean and Antarctic (annual) records show early onset of warming compared with the North Atlantic records (Fig. 1c, d, e).

Comparison with snapshot climate model simulations selected as part of an 'ensemble of opportunity' (Lunt et al., 2013) and presented in the most recent IPCC report (Masson-Delmotte et al., 2013) shows that the majority of models predict warmer than present conditions earlier than documented in the North Atlantic records (Fig. 2), while the magnitude of the reconstructed

early Southern Ocean and Antarctic warming is not captured (Fig. 2). An ensemble of LIG transient simulations with climate models of intermediate complexity or General Circulation Models (GCMs) with low resolution/accelerated forcing, also shows that only including orbital and GHG forcing results in peak Northern Hemisphere (NH) warming occurring earlier than that shown in the marine data records (Bakker et al., 2013). These results highlight not only the importance of producing defined time slices rather than a unique snapshot representative of the whole LIG but also that important missing processes in the

models are likely required to account for this temporal mismatch between data and model temperature anomalies (Capron et al., 2014). For example, previous GCM simulations did not consider freshwater forcing from melting of the NH ice-sheets prior and during the onset of the transition from glacial to interglacial conditions at 130 ka (Lunt et al., 2013). Accordingly, other work has invoked freshwater forcing from melting ice-sheets to account for a mismatch between model and data records

in the geological past (Smith and Gregory, 2009). Enhanced insolation forcing in the NH during the penultimate deglaciation resulted in rapid ice-sheet retreat and an increase in freshwater input to the North Atlantic and a suppression of the Atlantic Meridional Overturning Circulation (AMOC) near the end of the deglaciation (Carlson, 2008). In addition, marine sediment core evidence shows North Atlantic Deep Water (NADW) production was reduced compared with present day but recovered to present day values by 125 ka (Böhm et al., 2015; Lototskaya and Ganssen, 1999; Oppo et al., 1997).

A 130 ka climate model simulation (Holden et al., 2010), including freshwater forcing, shows warming over Antarctica with a freshwater input of 1 Sv into the North Atlantic between 50 and 70˚N, but still underestimates the temperature anomaly interpreted from East Antarctic ice cores. This mismatch between model and data is reconciled if the West Antarctic Ice-Sheet (WAIS) is removed in their simulation. However, a freshwater flux of 1 Sv is unrealistic for this time period when compared with rates of change in sea level (Grant et al., 2012, Figure 1h). Previous modeling studies (Loutre et al., 2014; Ritz et al.,

2011, Goelzer et al, in press) using climate models of intermediate complexity show a reduction in the strength of the AMOC as a result of freshwater input into the North Atlantic. Although Loutre et al. (2014) were able to model the delay in NH warmth in the early LIG when freshwater forcing was included, there is still a mismatch in timing and/or magnitude between their model temperature response and the temperature reconstructions. Govin et al. (2012) considered the melting of the Greenland ice-sheet and its influence on surface temperatures and NADW formation at 126 ka and showed a slow-down of

the AMOC along with reduced SSTs in the North Atlantic but the timing of the cooling from the new data synthesis of Capron et al. (2014) pre-dates conditions at 126 ka. Similar work by Bakker et al. (2012) and Otto-Bliesner et al. (2006) showed melting of the Greenland ice-sheet resulted in a reduction in the AMOC strength and cooling in the vicinity of the Labrador Sea. Goelzer et al (in press) showed that freshwater fluxes from the decaying Laurentide ice sheet during Termination II resulted in a decreased AMOC and associated increases in Southern Ocean temperatures, whereas freshwater from the

Antarctic ice sheet led to surface cooling in the same region. It is worth noting that mechanisms other than freshwater fluxes have been invoked which could cause millennial-scale variations in climate through changes in AMOC behavior. These include a salt oscillator in the North Atlantic (Peltier and Vettoretti, 2014), and wind stresses over the sub-polar gyre caused by changes in the Laurentide ice sheet geometry (Zhang et al., 2014). Furthermore, high latitude climates are influenced by changes in the mode of atmospheric circulation (e.g. Kleppin et al., 2015). However, our main focus here is on characterising

the role of freshwater fluxes in contributing to the LIG model-data mismatches.

The recent studies (Capron et al., 2014; Govin et al., 2012; Marino et al., 2015) based on proxy reconstructions of temperature and sea level speculated that the input of freshwater into the North Atlantic could explain the reconstructed NH versus Southern Hemisphere (SH) early LIG temperature pattern, via a bipolar response. Although previous modeling studies (e.g. Bakker et al., 2013; Holden et al., 2010; Loutre et al., 2014; Sanchez-Goni et al., 2012) have looked at the impact of freshwater forcing

on early LIG climate they did not link the response with the data reconstructions in the high latitude regions of the Northern and Southern Hemispheres and did not attribute this to a bipolar seesaw mechanism. As such, we perform the first rigorous model-data comparison approach to examine the impact and sensitivity of freshwater forcing on the high latitude climate of the early LIG to test whether the hypothesis of a bipolar mechanism is feasible in the framework of a comprehensive fully

coupled climate model to explain the difference in peak warmth conditions between hemispheres at 130 ka. We further perform an idealized simulation with the WAIS removed to test whether this has any additional influence on regional warming in our model framework, as recent work has indicated that some of the warmth seen in Antarctic ice core records during the LIG could partly be explained by a reduced West Antarctic ice sheet (Steig et al, 2015).

## 2 Experimental Design

In order to reconcile the high latitude mismatch between the data and model output at the beginning of the LIG for both hemispheres, we perform snapshot climate model simulations, representative of 130 ka conditions. The LIG starts at 129 ka when using a definition based on the eustatic sea level (Masson-Delmotte et al. 2013); however, considering dating uncertainties associated with paleoclimatic records during this time interval (see Govin et al. 2015 for a review), and the fact that defining the boundaries of interglacial periods is not trivial (see discussion in the Past Interglacials Working Group of

PAGES, 2016), we consider our 130 ka simulations as representative of the "early LIG". We use the UK Met Office fully coupled GCM, HadCM3 with an atmospheric horizontal grid spacing of 2.5˚ (latitude) by 3.75˚ (longitude) and an ocean horizontal grid spacing of 1.25˚ by 1.25˚ (Gordon et al., 2000), which includes the MOSES 2.1 land surface scheme where water and energy fluxes are calculated. For comparison with data we take advantage of the 130 ka data-based time slice produced by Capron et al. (2014). Compared with the pre-industrial period (see Table 1), the astronomical forcing resulted in

greater seasonality, leading to pronounced high northern latitude summer insolation during the early part of the LIG (Fig. 1a). GHG concentrations were similar to pre-industrial values based on records obtained from ice cores (Loulergue et al., 2008; Lüthi et al., 2008; Schilt et al., 2010) (Fig. 1b). In addition to prescribing these forcings we further vary the amounts of freshwater input between 0 and 1 Sv (Table 1, Figure 9) injected uniformly between 50 and 70˚N in the North Atlantic Ocean (Fig. 3) in order to determine the sensitivity of the model to freshwater forcing under an early LIG climate regime. Given the

uncertainty around the actual location of the freshwater flux, we prescribe an idealised hosing region. The climate simulations are run for 200 model years with fixed pre-industrial vegetation and ice-sheet distributions. According to the highly-resolved millennial-scale global sea level reconstruction based on Red Sea records (Grant et al., 2012) the rate of sea level rise was about 15.2 m/kyr during the glacial-interglacial transition (Fig. 1g,h). This is equivalent to a flux of approximately 0.17 Sv, an estimate in agreement with the 0.19 Sv calculated by Carlson (2008) based on coral records. As a consequence we choose a

NH freshwater input (assuming no contribution from the melting of the Antarctic ice-sheet at this time) of 0.2 Sv (HadCM3_BRIS_130ka_0.2Sv) as our best-estimate scenario with which to compare our model temperature output and the high latitude data synthesis at 130 ka. We also perform a 130 ka simulation forced with a freshwater forcing of 0.2 Sv and the

WAIS removed and its bedrock after removal defined to be 200 m above sea level (HadCM3_BRIS_130ka_0.2Sv_NOWAIS) and replaced with a bare soil surface, more akin to what is observed in the Dry Valleys today. A land surface type was chosen instead of ocean due to instabilities in the ocean numerics in HadCM3 close to the South Pole. However, Holden et al. (2010) show with the GENIE climate model that replacing the WAIS with ocean rather than land results in only a slight increase in

the surface air temperatures over Antarctica. Given the uncertainty in the location and rate of freshwater forcing associated with the WAIS removal, we do not prescribe additional freshwater fluxes from the WAIS. Finally, we also perform a 130 ka simulation forced with a freshwater forcing of 0.2 Sv and the WAIS removed, but with WAIS replaced with shrubs instead of bare soil, to test the response to uncertainty in the land-cover type which would replace the ice sheet. We perform analysis on the last 50 model years of each simulation. To test the robustness of the results to the 200-year simulation length, we extended

the 130 ka simulation with 0.2 Sv of freshwater forcing for a further 200 model years (400 years in total). In the Southern Ocean the rate of change of summer-SST with time is very small, and the difference between the 50-yr climate mean JFM anomaly after 200 years compared with the 50-yr climate mean after 400 years is very small (not shown); the difference ranges between -0.5 and 0.5°C for the majority of the region, which is well within the uncertainty of the data synthesis from Capron et al. (2014) of 2.6°C on average.

For the model-data comparison, two methods have been used to calculate the Root Mean Square Error (RMSE) to determine the influence of clustering of the data points on the RMSE calculation. Method 1 is based on comparing each observation ($x_i$) at 130 ka with its coincident grid cell model value ($y_i$) according to Eq. (1):

$$RMSE_1 = \sqrt{\frac{\sum_{i=1}^{N}(x_i - y_i)^2}{N}}, \qquad\qquad (1)$$

where $N$ is the total number of observations. Method 2 takes into account the effect of clustering of the observations when
compared with model values. The RMSE is calculated according to Eq. (2):

$$RMSE_2 = \sqrt{\sum_{i=1}^{G}\frac{\left(\sum_{j=1}^{n_i}|x_{ij} - y_{ij}|/n_i\right)^2}{G}}, \qquad\qquad (2)$$

where $G$ is the total number of groups of clustered data points, $n_i$ is the number of observations in each group, $x_i$ is the observation and $y_i$ is the model value. Each group (chosen based on geographical proximity) is shown in Fig. 5 according to a different color for the data compilation at 130 ka and 125 ka for the three geographical regions considered. The absolute error
is calculated between each observation and its coincident model value then averaged over the group

## 3 Results and Discussion

Figure 4a shows results from the 130 ka climate simulation with no additional freshwater input compared with the Turney and Jones (2010) time slice, the latter assuming synchronous temperature changes across the globe during the LIG. Figure 4b shows a comparison with the high-latitude 130 ka time slice from the Capron et al. (2014) synthesis. Note that Turney and
Jones (2010) interpret the records as annual temperature means while Capron et al. (2014) interpret the marine records as

summer temperature means, as proposed by the authors of the original papers, and the ice core records as annual means. In the North Atlantic, any similarity between the model and the Turney and Jones data is misleading as the LIG temperature maximum recorded by their study generally occurred later than 130 ka; a similar compilation restricted to data from 130 ka would be much colder than the data shown in Fig. 4a. This behavior is seen in the Capron et al. (2014) 130 ka data synthesis,

now interpreted as seasonal, showing a cooling in the North Atlantic. The model simulation with only orbital and GHG forcing (Fig. 4b) matches poorly to the 130 ka compilation of Capron et al. (2014), with too high temperature anomalies in the NH ($RMSE_1$ = 5.9°C) and too low temperature anomalies in the SH ($RMSE_1$ = 2.4°C). The mismatches in temperature between data and model are much too large to be resolved even taking into account the uncertainties on the marine temperature reconstructions. Regarding temperatures over Antarctica, near-surface annual air temperature anomalies are several degrees

cooler in the model compared with the Capron et al. (2014) synthesis, even considering the uncertainty in the temperature reconstructions ($RMSE_1$ = 1.7°C). Furthermore, Table 2 shows that the RMSE result for each region is similar for both $RMSE_1$ and $RMSE_2$. Note that the Capron et al (2014) dataset cites uncertainties in the data of 2.6ºC on average for the data, which we do not propagate into the RMSE values.

Inclusion of a constant freshwater forcing of 0.2 Sv in the North Atlantic in the model results in a decrease in the strength of

the AMOC of more than 10 Sv, and an associated change from warming in the North Atlantic to a cooling compared with present day (Fig. 4c). This leads to a considerable improvement in $RMSE_1$ from 5.9°C to 3.3°C for the North Atlantic compared with Capron et al. (2014). A warming compared with present is observed in the climate model during the summer months for the Southern Ocean, similar to when no freshwater forcing is included, but is more extensive in the vicinity of the WAIS with SSTs up to 2°C warmer than present. However, there is a lack of temperature records from ocean sediment cores to further

validate the model simulation in this region. The addition of freshwater into the North Atlantic results in a bipolar seesaw response (Stocker, 1998) with a redistribution of heat between the hemispheres resulting from decreased northward heat transport through the Atlantic (Crowley, 1992), although the response in the NH is stronger compared with that simulated in the Southern Ocean. Here we use a snapshot approach and, therefore, do not consider the timing of phasing between the hemispheres with relation to the bipolar seesaw.

Other mechanisms have been suggested to explain the colder than present North Atlantic at 130 ka. A study using the NorESM climate model (Langebroek and Nisancioglu, 2014) (see Fig. 2 and Table 2) shows cooling in the North Atlantic without the need to invoke freshwater input. They attribute this to an expansion of the southeastern part of the subpolar gyre and an eastward shift in the North Atlantic Current combined with a stronger AMOC. However, marine sediment core evidence suggests that the AMOC was temporarily weaker at this time (e.g. Böhm et al., 2015). Furthermore, this cooling persists at

125 ka when the data shows an overall warming compared with present day (see Fig. 6 and Table 2 for details).

The Southern Ocean warming is coherent with the warmer-than-present conditions suggested in ice core records from East Antarctica. There is a small improvement in the RMSE over East Antarctica ($RMSE_1$ = 1.5°C) when freshwater forcing is included compared to without ($RMSE_1$ = 1.7°C), although the model is still too cold by up to 2°C, similar to Holden et al. (2010). Recent work has suggested that the southern hemisphere cooling arising from changes in the northward heat transport

in the Atlantic, such as we have here, can be communicated to Antarctica by feedbacks associated with sea ice expansion; in particular, the expanded seaice reduces the winter warming effect of the Southern Ocean (Pedro et al, 2016).

Although the new LIG data synthesis of Capron et al. (2014) does not extend to continental records and to latitudes lower than 45°N, note that forcing the model at 130 ka with a 0.2 Sv freshwater flux leads to simulated surface air temperatures over Europe that are consistent with existing datasets (e.g. Sanchez-Goni et al., 2012; see Fig. 7).

The contribution of the Greenland ice-sheet to global LIG sea level rise has recently been quantified (Born and Nisancioglu, 2012; Colville et al., 2011; Helsen et al., 2013; NEEM community members, 2013; Quiquet et al., 2013; Stone et al., 2013), with the IPCC Fifth Assessment Report stating a range very likely between 1.4 and 4.3 m of equivalent sea level height (Masson-Delmotte et al., 2013). Taking contributions from thermal expansion and mountain glaciers into account and that global sea level was at least 6 m higher than today (Dutton et al., 2015) this implies that a contribution is likely also required from the WAIS (noted specifically by Colville et al. (2011)), and/or other parts of the Antarctic Ice-Sheet. Although studies have suggested the possibility of an East Antarctic contribution (Bradley et al., 2012; Fogwill et al., 2014; Pingree et al., 2011) this has yet to be quantitatively supported by observational or modeling evidence. Future research using ice-sheet models could investigate whether the warming of the Southern Ocean via the bipolar seesaw mechanism leads to enhanced basal melting of the WAIS and retreat of the grounding line (Joughin et al., 2012; Timmermann and Hellmer, 2013, Goelzer et al, in review, DeConto and Pollard, 2016) at the beginning of the LIG. Indeed, a recent study has suggested that the water isotopic data from the Mount Moulton ice core drilled in West Antarctica compared with water isotopic profiles from East Antarctic ice cores, is consistent with a collapse of the WAIS during the LIG (Steig et al., 2015). This potential melting of the WAIS during the early LIG could explain or partially explain the mismatch between the model simulations and Southern Ocean/East Antarctic data timeslices at 130 ka.

However, in the additional simulation (Fig. 8a) where we remove the WAIS and include the freshwater forcing input of 0.2 Sv (HadCM3_BRIS_130ka_0.2Sv_NOWAIS), the model-data match is not improved (see Table 2) over East Antarctica and still underestimates the temperature response by at least 1˚C (Fig. 8a), although there is an increase in overall warming compared with the case when only freshwater forcing is considered (Fig. 8b). This result supports, to some extent, the findings of Holden et al. (2010) where the WAIS was removed and 1 Sv of freshwater was added in the North Atlantic leading to enhanced warming over East Antarctica but, in our case a more realistic amount of freshwater forcing based on data is implemented. There is some uncertainty as to the extent or type of vegetation which may or may not have grown on an unglaciated West Antarctica during the LIG, and the vegetation type replacing a previously glaciated surface can have a significant effect on the magnitude of warming (Stone and Lunt, 2013). Figure 8c shows that warming over Antarctica is sensitive to the land surface type chosen to replace the WAIS with an increase in annual temperature by up to 2˚C over Antarctica when covered with a shrub surface type instead of bare soil. Another study using the CCSM3 model (Otto-Bliesner et al., 2013), but without additional NH freshwater forcing, found very limited improvement in the model response when the WAIS was removed and replaced with ocean. It is possible that our simulations with WAIS replaced by a land type is overestimating the warming.

Threshold behavior of modeled AMOC strength in response to varying amounts of freshwater forcing has been previously investigated with models showing a range from 0.1 to 0.5 Sv at which NADW formation can no longer be sustained (Rahmstorf et al., 2005). As a result of this range in response of AMOC collapse to freshwater input, we perform an analysis of the response of the high latitude regions to varying amounts of freshwater forcing in the North Atlantic to test the sensitivity of the model

under 130 ka forcing conditions. This is similar to the study of Bakker et al. (2012) which looked at the sensitivity of the AMOC to Greenland ice-sheet melt during the LIG using a climate model of intermediate complexity. Figure 9 shows the model summer North Atlantic temperature response (averaged over the locations for which Capron et al. (2014) provide temperature records) for freshwater input varying from 0 to 1 Sv compared with the average NH temperature anomaly from the Capron et al. (2014) dataset (horizontal dashed line). In addition the strength of the AMOC at 30˚N for the varying amounts

of freshwater forcing is included. It is clear that HadCM3 shows a distinct threshold at around 0.2 Sv under LIG boundary conditions where freshwater input amounts greater or equal to this lead to sufficient freshening in areas of NADW formation and a reduction in the mixed layer depth in these regions. This freshening results in reducing the overturning strength of the AMOC considerably by more than 10 Sv. As a result the average temperature response observed in the North Atlantic becomes cooler than present due to a reduction in northward ocean heat transport. The weakening of the overturning circulation occurs

within 50 model years.

The implication of these simulations and the NH forcings depicted in Fig. 1 is that the freshwater forcing from the melting of the remnant ice-sheets provides a mechanism to warm the Antarctic and Southern Ocean during the early LIG for a limited amount of time. From about 128 ka onwards, NH surface temperature records and modeling studies (Capron et al., 2014) show surface warming relative to today also occurred in the NH. At 125 ka, when the meltwater flux (Fig. 1) had likely returned to

a low baseline, the match between HadCM3 (orbital and GHG forcing only) and a similar compilation of data (targeted at 125 ka, Fig. 6) is reasonable (Capron et al., 2014), strengthening the case that a bipolar seesaw signal is required to reconcile the evolution of temperature between 130 and 125 ka. This inter-hemispheric bipolar seesaw pattern in temperature response during the penultimate deglaciation first suggested by CLIMAP Project Members (1984) was also highlighted in recent studies by Masson-Delmotte et al. (2010) and Marino et al. (2015) while such a pattern has also been shown during Termination 1

(Shakun et al., 2012). Thus, this hemispheric asynchrony represents an important feature of at least the last two glacial terminations. However, in order to fully explore the temporal variations in temperature through the LIG, fully transient simulations with time-evolving forcings would be required.

## 4 Conclusions

Using new 130 ka snapshot GCM simulations and benefiting from the advent of a new time-varying data-based representation

of the climate evolution across the LIG, we provide valuable modeling insights to explain the inter-hemispheric asynchrony in temperature response during the early part of the LIG. We show that inclusion of freshwater input (determined from data records) into the North Atlantic due to the melting of the remnant NH ice-sheets from the penultimate glaciation can explain

the cold summer temperature anomalies observed in the NH paleorecords and an extensive early warming of the SH at 130 ka. Conversely, removing the WAIS in the simulations does not improve the model-data comparison in East Antarctica or the Southern Ocean. However, the lack of data coverage does not allow us to draw conclusions regarding the configuration of the LIG WAIS. Our new results highlight the need for additional paleoclimatic records (e.g. marine sediment records in the vicinity of the WAIS) in order to better characterize both the spatial and temporal high latitude climatic patterns during the LIG. Possible future work should include analyzing ice-sheet model simulations of the WAIS to test whether the ocean warming in these simulations is substantial enough to increase basal melting of the ice-sheet and grounding line retreat, and to account for the warming observed from ice core records in East Antarctica at 130 ka. This study shows the importance of studying the LIG not in isolation but also in the context of the preceding glaciation. It further emphasizes the importance of considering other forcings in addition to changes in orbital and GHG forcings (which can lead to abrupt changes in the climate) in future model simulations to improve the evaluation of their impact on climate change, particularly in the high latitude regions.

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

**Acknowledgements**

The model data for this paper is available on request from the following website: http://www.bridge.bris.ac.uk/resources/simulations. The data synthesis can be accessed from the online Supplementary Material from Capron et al. (2014). This work was carried with funding from the UK-NERC consortium iGlass (NE/I009906/1) and is also a contribution to the European Union's Seventh Framework programme (FP7/2007–2013) under grant agreement

243908, "Past4Future. Climate change – Learning from the past climate". This is Past4Future contribution no 85. The climate model simulations were carried out using the computational facilities of the Advanced Computing Research Centre, University of Bristol - http://www.bris.ac.uk/acrc/. We thank the reviewers whose constructive comments improved the paper.

25

| | **Greenhouse Gas concentration** | | | **Orbital parameters** | | |
|---|---|---|---|---|---|---|
| | $CO_2$ (ppm) | $CH_4$ (ppb) | $N_2O$ (ppb) | Obliquity (˚) | Eccentricity | Perihelion (day of year) |
| | 257 | 512 | 239 | 24.25 | 0.0401 | 121.8 |
| **Freshwater forcing (Sv)** | 0.0 | 0.1 | 0.2 | 0.3 | 0.4 | 0.5 | 1.0 |

| WAIS; GrIS state | M; M | M; M | 1. M; M <br> 2. N; M | M; M | M; M | M; M | M; M |
|---|---|---|---|---|---|---|---|

**Table 1.** Greenhouse gas concentrations, orbital and freshwater forcing, state of the ice sheets (GrIS-Greenland Ice-Sheet; M = modern day ice sheet; N=No ice-sheet and orography flattened) for the GCM simulations at 130 ka.

| | NH SST (JAS) | SH SST (JFM) | EAIS (ANN) |
|---|---|---|---|
| **CCSM3_NCAR_130ka** | 7.8 (6.6) | 2.6 (1.9) | 2.0 (2.0) |
| **COSMOS_AWI_130ka** | 5.1 (4.6) | 2.6 (1.8) | 1.5 (1.4) |
| **CSIRO_UNSW_130ka** | 4.5 (4.0) | 2.7 (1.9) | 2.4 (2.4) |

| | | | |
|---|---|---|---|
| **NORESM_BCCR_130ka** | 3.8 (3.3) | 2.3 (1.5) | 1.7 (1.6) |
| **HadCM3_BRIS_130ka** | 5.9 (5.6) | 2.4 (1.8) | 1.7 (1.6) |
| **HadCM3_BRIS_130ka_0.2Sv** | 3.3 (2.8) | 2.1 (1.5) | 1.5 (1.4) |
| **HadCM3_BRIS_130ka_0.2Sv_NOWAIS** | 3.1 (2.7) | 2.3 (1.8) | 1.5 (1.4) |
| **HadCM3_BRIS_125ka** | 3.5 (3.7) | 2.3(1.7) | 0.8 (0.7) |
| **NORESM_BCCR_125ka** | 3.1 (2.6) | 2.2 (1.6) | 1.1 (1.1) |

**Table 2**. Root Mean Squared Error (RMSE) for NH and SH SSTs and EAIS near surface air temperature regions. RMSE is calculated according to Eq. (1) (*RMSE*$_1$) and Eq. (2) (*RMSE*$_2$, values in brackets). The model output is compared with the 130 ka and 125 ka time slices from Capron et al. (2014). Note that the simulations without freshwater forcing included were previous described in Lunt et al. (2013) and references therein.

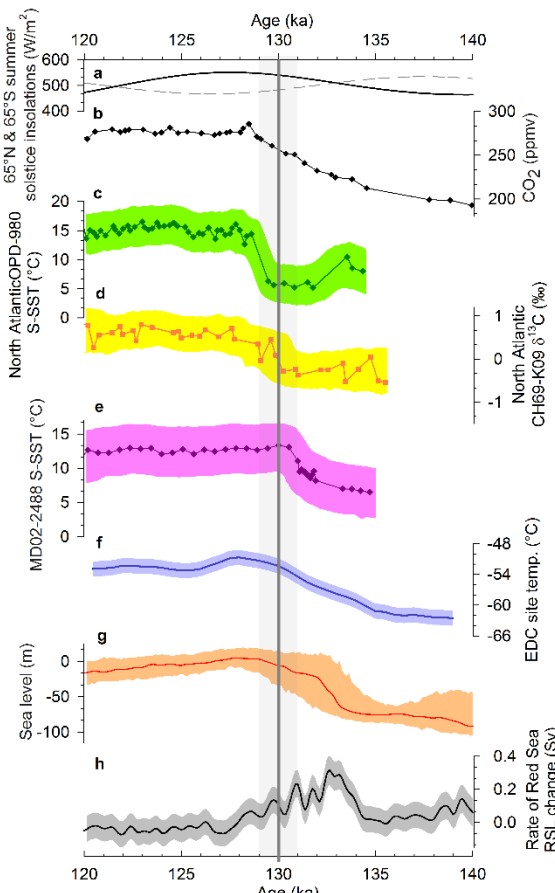

**Figure 1**. **(a)** 65°N (black) and 65˚S (grey) summer insolation (Laskar et al., 2004). **(b)** EDC ice core $CO_2$ concentration (Schneider et al., 2013) (black). **(c)** North Atlantic core ODP-980 summer-SST reconstruction (Oppo et al., 2006) (green) and associated $2\sigma$ uncertainty envelope (Capron et al., 2014) (light green). **(d)** North Atlantic core CH69-K09 $\delta^{13}C$ record (Govin et al. 2012 and $2\sigma$ uncertainty envelope (this study, see details on the methodology in Capron et al., 2014). **(e)** Southern Ocean core MD02-2488 summer-SST reconstruction (Govin et al., 2012) (pink) and $2\sigma$ uncertainty envelope (Capron et al., 2014) (light pink). **(f)** EDC surface temperature reconstruction (dark blue) and associated 1.5˚C uncertainty envelope (Masson-Delmotte et al., 2011) (light blue). Note that Govin et al. (2015) reports in the Table 5 of their paper that the Antarctic reconstructed surface temperature (based on EDC dD) starts increasing at 135.6 ±2.5 ka based on the use of the RAMPFIT software. **(g)** Red Sea relative sea level (RSL) data (probability maximum, red) with 95% confidence interval (Grant et al., 2012) (orange). **(h)** Red Sea rate of RSL change (probability maximum for the first-order time derivative, in Sv, black) with 95% confidence interval (Grant et al., 2012) (grey). Note that ice and marine records from **(a)** to **(f)** are shown on the AICC2012 ice core chronology (Bazin et al., 2013; Capron et al., 2014; Veres et al., 2013) while the Red Sea records **(g, h)** are displayed on their original age scale which is independent from the AICC2012 ice core chronology (see Grant et al., 2012 for details) The grey vertical line marks 130 ka. The grey band highlights the 129-131 ka time interval that has been considered for the construction of the 130 ka data based time slice for surface temperature (see Capron et al. 2014 for details on the methodology).

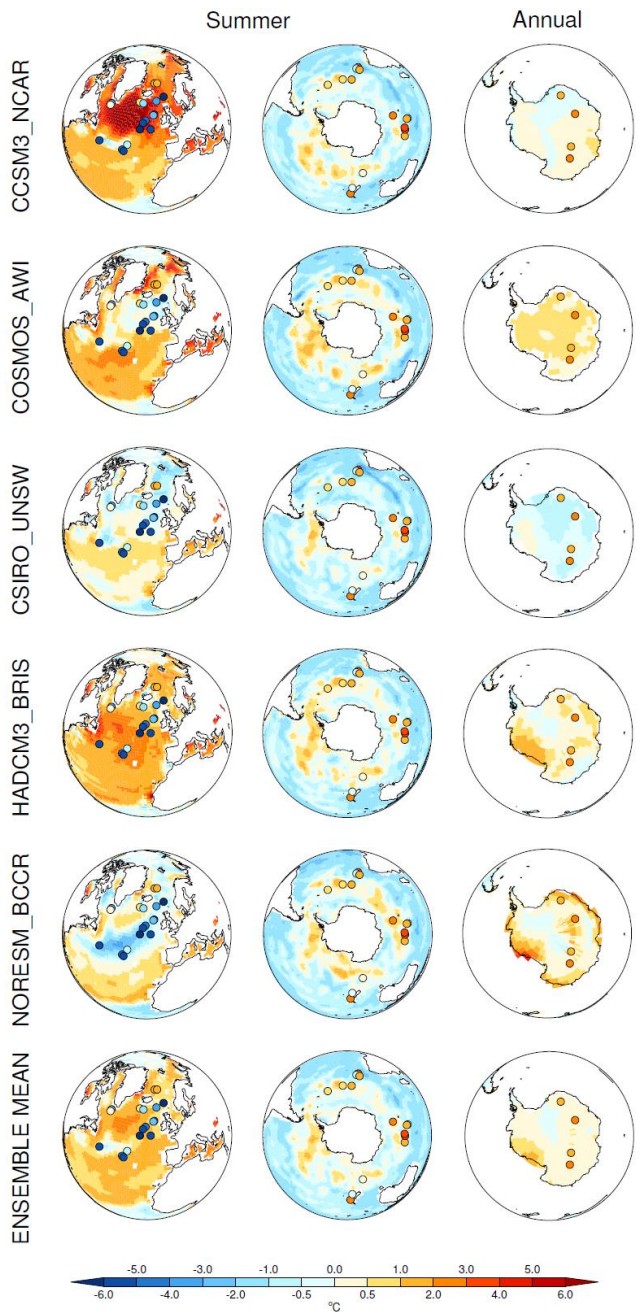

**Figure 2.** Simulated summer (NH-July, August, September; SH-January, February March) (left and middle panels) SST and annual (right panel) surface air temperature change relative to preindustrial, for GCM model results previously published (Lunt et al., 2013) and their ensemble mean. The simulations are compared with the new data-based 130 ka time slice (Capron et al., 2014).

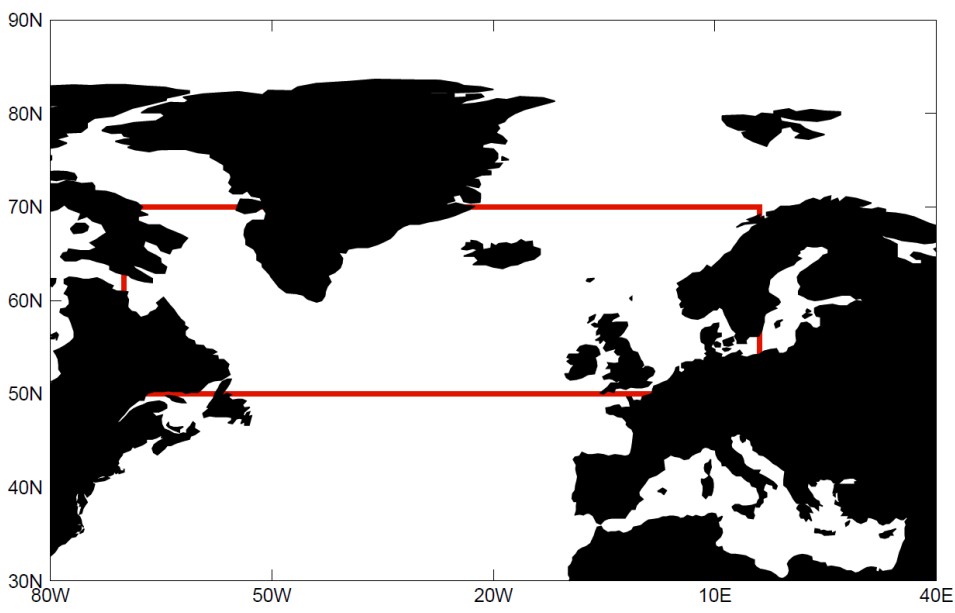

5 **Figure 3**. The North Atlantic region for freshwater input denoted by the red box (50-70˚N). Note that the freshwater amount is evenly distributed within this region.

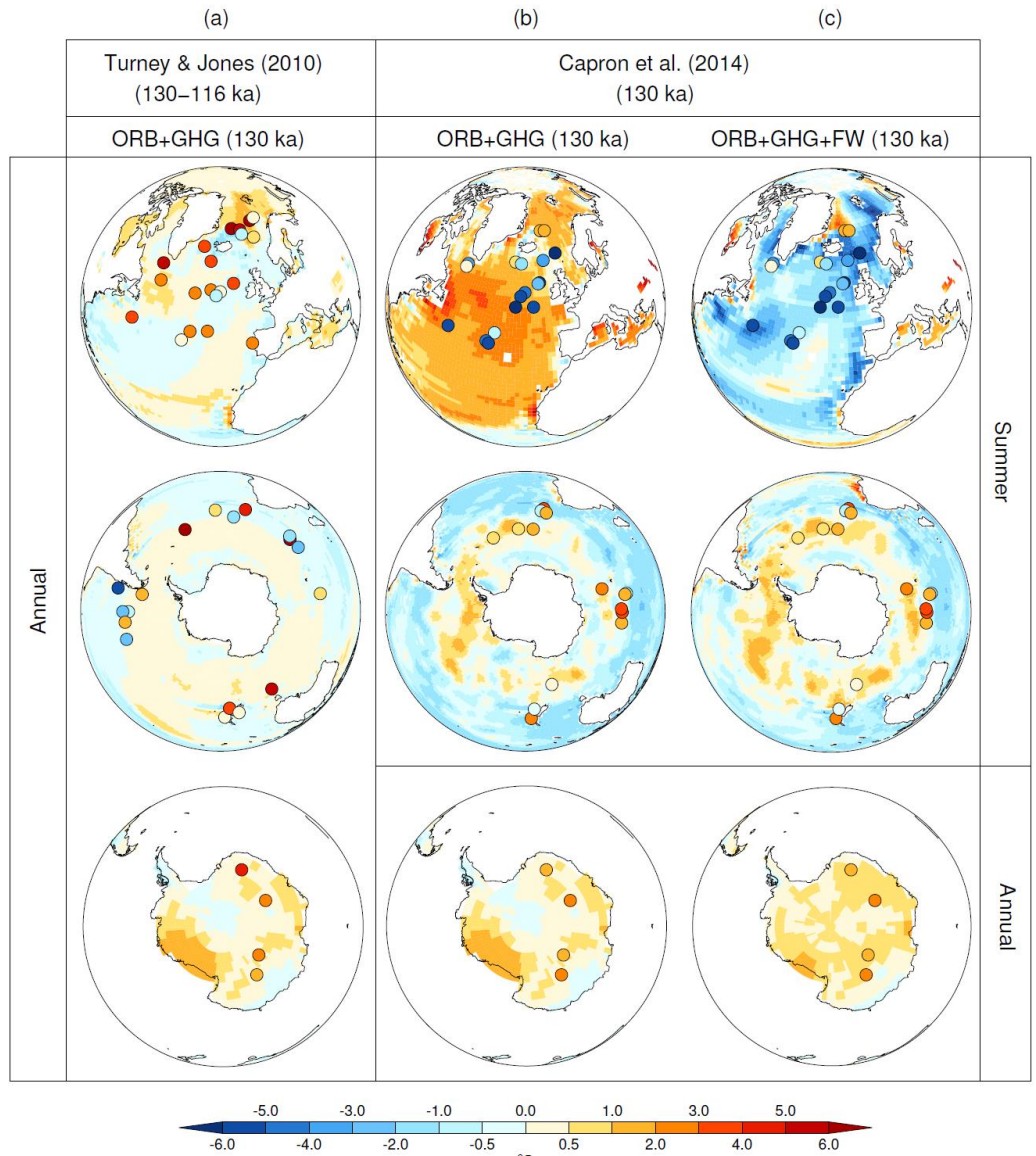

**Figure 4.** Simulated 130 ka SSTs and near surface air temperature anomalies compared with data for the high latitude regions. The top two rows are SSTs (annual or summer as labelled) and the bottom row is annual mean near surface air temperature. Left panel **(a)** LIG peak warmth data synthesis of Turney and Jones (2010) (dots) compared with 130 ka annual temperature anomalies (GHG and orbital forcing only). Middle panel **(b)** The 130 ka data-based time slice (dots) compared with simulated summer-SST anomalies for the NH (July, August, and September) and SH (January, February, and March) (GHG and orbital forcing only). Right panel **(c)** The 130 ka data-based time slice (dots) compared with summer-SST anomalies for the NH and SH (GHG, orbital forcing and a constant freshwater input of 0.2 Sv into the North Atlantic). Note the non-linear temperature scale. Anomalies calculated relative to the preindustrial for the model and relative to modern for the data.

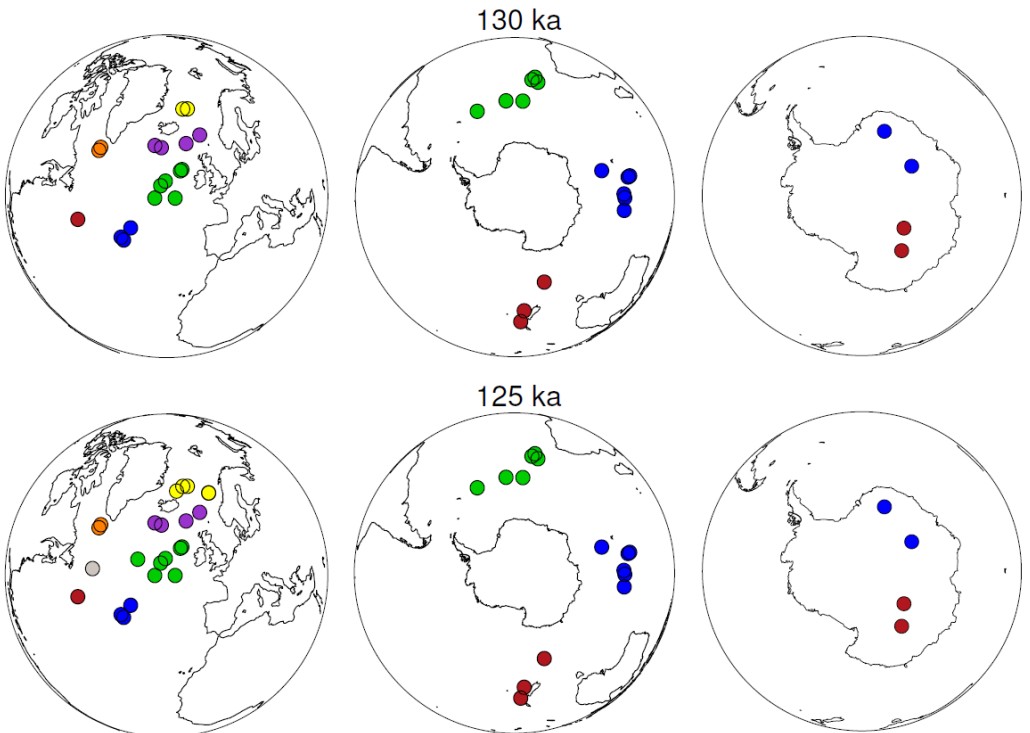

**Figure 5.** Locations of 130 ka and 125 ka data-based time slice data from Capron et al. (2014). The colors denote the groups of data used in Method 2 (Eq. (2)) to calculate the RMSE for each region.

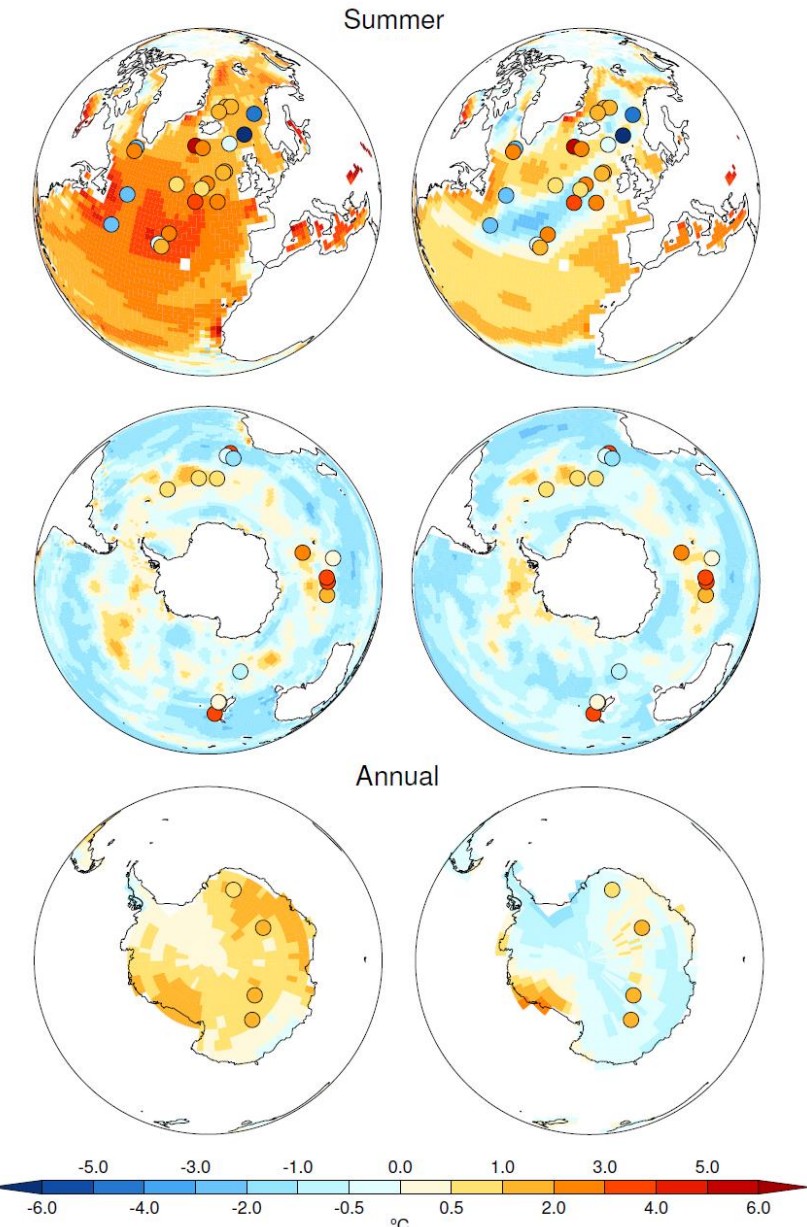

**Figure 6.** Simulated summer (NH-July, August, September; SH-January, February March) SST and annual (bottom panel) surface air temperature anomalies at 125 ka compared with the Capron et al. (2014) 125 ka data-based time slice. Left panel: HadCM3, right panel: NorESM. Table 2 shows a similar agreement with data for NorESM and HadCM3 (which has no cooling in the North Atlantic) in the NH when compared with the 125 ka data-based time slice from Capron et al. [2014]. However, NorESM shows poor agreement with the data synthesis over Antarctica. Anomalies calculated relative to the preindustrial for the model and relative to xxx for the data.

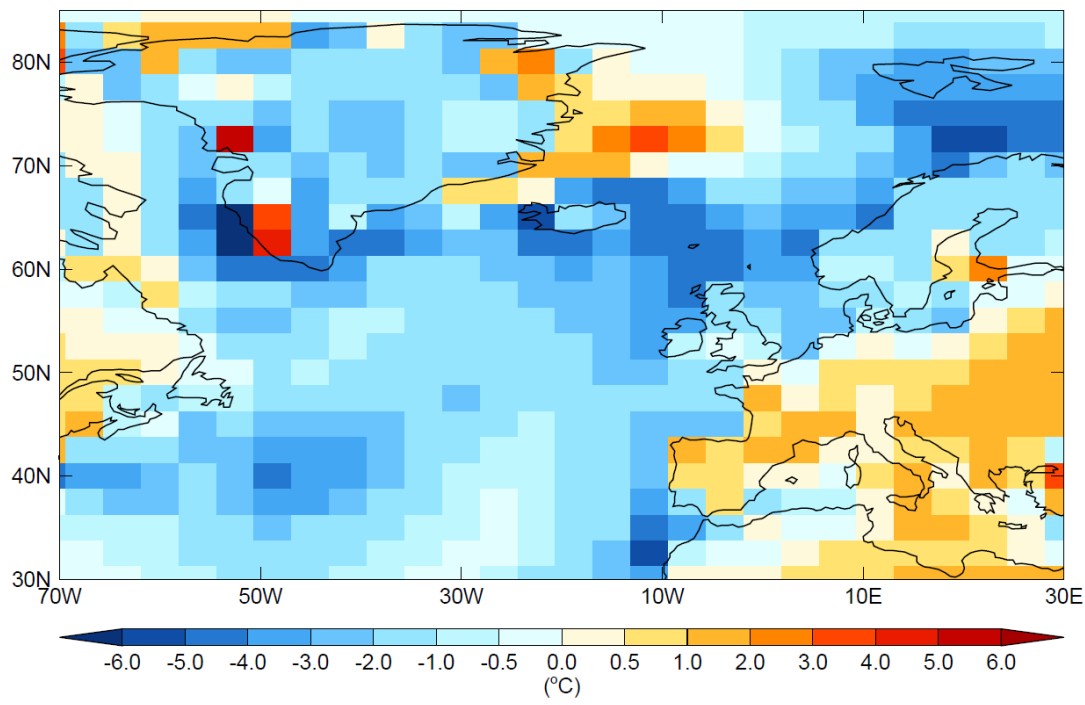

5  **Figure 7**. Simulated summer (July, August and September) near surface air temperature anomaly compared with preindustrial at 130 ka over southern Europe and the North Atlantic region. The simulation is forced with 0.2 Sv of freshwater flux as well as changes to the GHGs and orbital forcing (HadCM3_BRIS_130ka_0.2Sv).

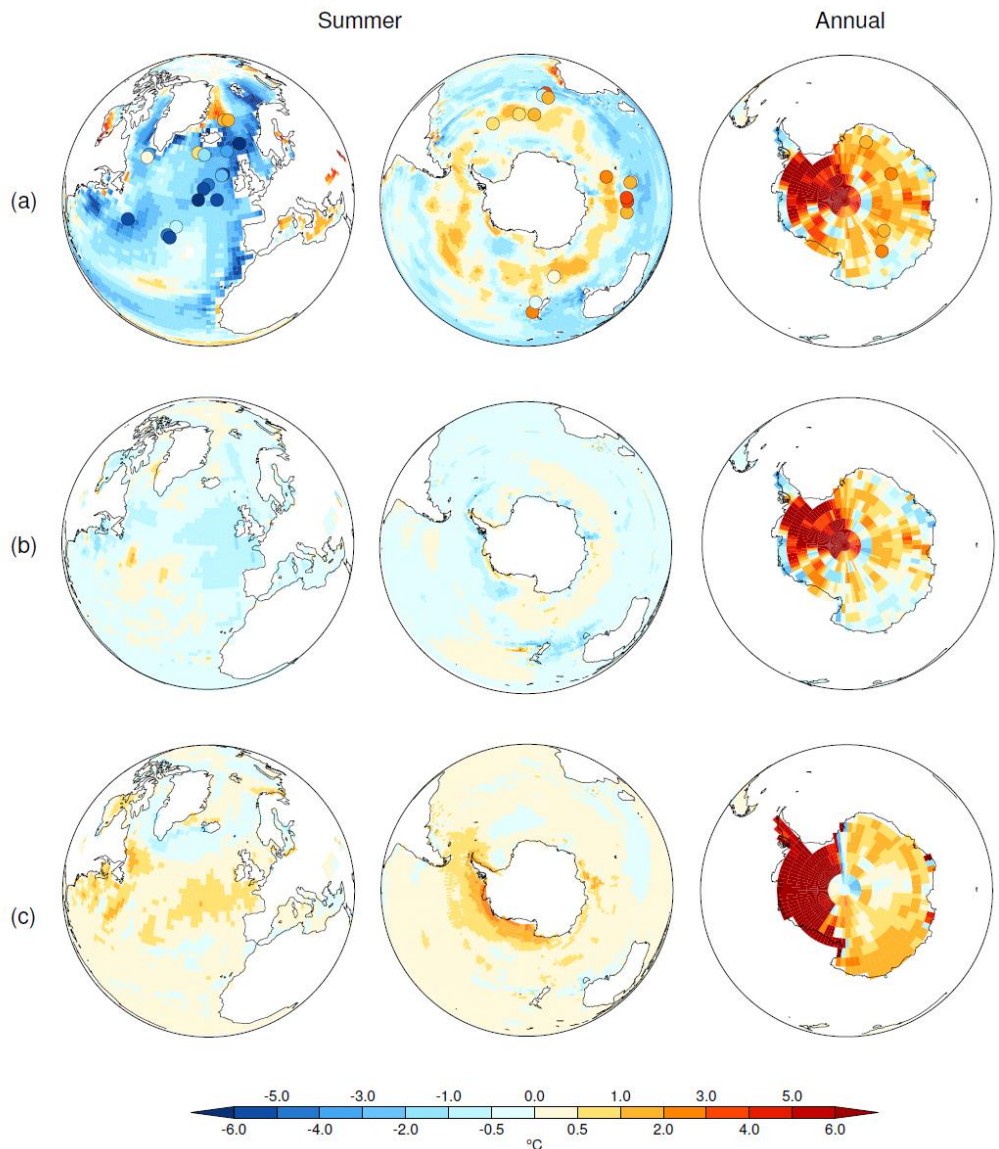

**Figure 8. (a)** Simulated 130 ka SST and near surface air temperature anomalies, with the WAIS removed and replaced with bare soil and a North Atlantic freshwater input forcing of 0.2 Sv, compared with the Capron et al. (2014) 130 ka time slice. **(b)** Difference in SST and near surface air temperature between (a) and the 130 ka simulation with 0.2 Sv freshwater forcing only (Fig. 4c). **(c)** Difference in SST and near surface air temperature when the WAIS is replaced with shrub compared with bare soil (Fig. 8a). Left: summer-SST anomalies for the NH (July, August, and September); middle: summer-SST anomalies in the SH (January, February, and March); right: annual near surface air temperature anomalies over Antarctica. It has been shown that this warming over East Antarctica is attributed to climatic effects rather than isostatic effects arising from a reduction in the WAIS (Bradley et al., 2012).

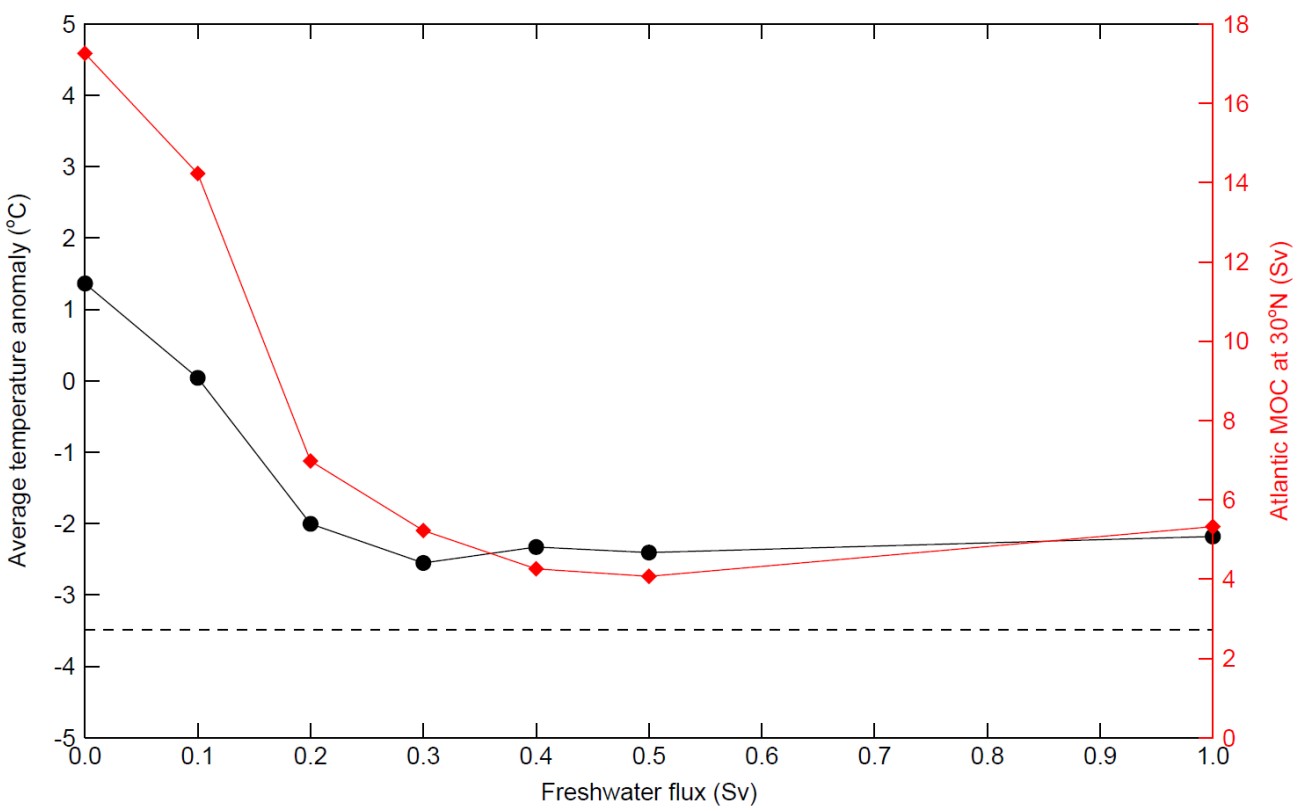

**Figure 9.** Response of the North Atlantic summer-SSTs and strength of the overturning circulation to varying amounts of freshwater input injected between 50 and 70°N (0 Sv, 0.1 Sv, 0.2 Sv, 0.3 Sv, 0.4 Sv, 0.5 Sv, 1 Sv). Left axis: Average 50-year model temperature anomaly relative to present day. The temperature is averaged over all model grid-boxes where data points are located. Right axis: 50-year average maximum Atlantic Meridional Overturning Circulation strength at 30°N. The dashed horizontal line corresponds to the average summer (July, August, September) temperature anomaly for all marine data located at >40°N for the 130 ka time slice (Capron et al., 2014).