# Peer review of "Impact of melt water on high latitude early Last Interglacial climate"

_Climate of the Past, 2016_

## Referee Comment (RC1) · Anonymous Referee #1 · 12 Feb 2016

Stone et al. present an interesting and thorough model-data comparison focusing on the issues surrounding the early Last Interglacial climate evolution: Why were the Southern Ocean and Antarctica warm when the North Atlantic still appear to have been relatively cold compared to full interglacial conditions? And what were the potential roles of meltwater from remnant ice sheets, the Atlantic Meridional Overturning Circulation and the West Antarctic Ice Sheet? It is a good study and well structured paper that is well suited for publication in Climate of the Past. Below I've listed a number of comments that I think should be included to clarify and improve the manuscript.

Main comments:

An important contribution and the most novel part of this work, is the simulations in which both a NH meltwater source and changes in the WAIS are simulated. Currently

this topic is only introduced in the very last line of the introduction and the simulations are not even mentioned in the conclusions. I think this part of the manuscript should be discussed more comprehensively in the introduction, including previous work (e.g. Goelzer et al, 2016, Steig et al., 2015) and indications that such a WAIS collapse has occurred during or previous to the LIG. In the conclusions question could be addressed such as 'What did the WAIS simulations tell us?', or 'How should future simulations on this topic be improved in order to better address the issues discussed in the manuscript'?

Minor comments:

Page 1 line 17: As discussed in the manuscript, a number of previous studies used a model-data approach to investigate the impact of Northern Hemisphere freshwater forcing on the early LIG climate, so perhaps 'for the first time' is a little too strong, although this study is certainly more thorough and presents, as is mentioned, a more 'integrated model-data approach'.

Page 1 line 22: If the LIG is from 129-116ka, can one still consider 130ka as early LIG? Perhaps a technical detail, but on the other hand a good illustration of the broader issue that defining deglacial and interglacial periods is not trivial and perhaps not even desirable.

Page 2 line 14: What is meant with 'partially account for changes in seasonality of precipitation'? Do the uncertainty estimates account for this or can part of the ice core oxygen isotope changes be accounted for by changes in the seasonality of precipitation?

Page 3 lines 7-9: It is mentioned that Loutre et al. (2014) already performed a model-data comparison including NH freshwater fluxes for the early LIG, but that their work still showed model-data mismatches. Doesn't the present work still show these? Perhaps they became smaller? Or we have a better understanding of why these mismatches occur?

[Figure]

Page 3 lines 16-19: Both Bakker et al. (2013) and Loutre et al. (2014) included a model-data comparison, be it small and less rigorous than the one presented here. Another such model-data comparison for this time interval was performed by Sanchez-Goni et al. (2012).

Introduction: The recent work by Goelzer et al. (2016) should be discussed as well since it is closely related to the questions that are addressed in this manuscript.

Experimental design: Discuss some aspects of the experimental design in a little more detail: Are 200 year simulations are sufficiently long to investigate a bi-polar seesaw response? Hosing a large region between 50-70N seems highly idealized. What do we know about the distribution of meltwater during that period and what difference could it make to include a more realistic meltwater scheme? Meltwater from the WAIS is neglected. Why and how could this impact the results?

Page 4 lines 28-30: What about model uncertainties or inter-model differences in simulated temperature anomalies, can those explain the model-data mismatch?

Page 6 lines 4-16: It does not really become apparent from this paragraph that another important reason to perform simulation in which the WAIS is removed is because this could explain the persisting SH model-data mismatch.

Page 6 line 18: From table 2 it appears to me that the number are identical so why is the model-data match slightly improved?

Page 6 line 24: Why would you replace it with shrubs? Is there any indications that those would grow there during the LIG? And related to that, why is such a big impact found between replacing it with bare ground or shrubs, I would expect that that region is covered with snow year round?

From the last paragraph of the results section and figures 4 and 6 it is not fully clear to me how SH temperatures evolved during the LIG and how this relates to the limited NH freshwater forcing after 127ka. This scenario would suggest that after the early LIG,

when the NH freshwater returned to a low baseline, the bi-polar seesaw seized, potentially leading to cooling in the SH. Is that seen in the 125ka time-slice of Capron et al. (2014)? If not, how could this be explained? Please discuss this very interesting topic in more detail in this paragraph and perhaps include suggestions for future research on this topic.

Table 2: The improvement of the SH and EAIS model-data match when included the NH 0.2Sv meltwater forcing is surprisingly small. What are we missing?

Table 2: The lowest two lines (125ka) are they also compared with the 125ka time-slice of Capron et al. (2014)? Please explain in the caption.

Table 2: Mention in the caption which simulations were previously published and which are newly performed for this study.

Figure 1: Consider including a proxy records showing the AMOC evolution during this period. For instance one of the d13C records shown by Sanchez-Goni et al. (2012) and Govin et al. (2012).

Figure 1: An additional vertical axis showing the rate of sea level change in Sv would be easier to compare with for instance figure 9.

Figure 1: the 'early SH warmth' is not very clear in EDC temperatures. Please clarify.

Figure 1: please include in the caption a description of the grey band shown in the figure.

Figure 3: This could perhaps also be shown in another figure that shows a map of the North Atlantic region.

Figure 6: Why is the model response so different over Antarctica while it is so similar over the Southern Ocean? Is seems unrelated to the changes in the North Atlantic. Is this difference also there at 130ka?

Figure 8: Why does the North Atlantic show a warming in figure c?

Technical comments:

Page 1 lines 13-17. Line is very long and difficult to read. Please rewrite.

Page 1 line 22 and 27: At multiple locations double brackets are used, either like (...(..)) or like (...)(..). Consider adjusting.

Page 2 line 2: consider removing 'build'.

Page 2 line 5: consider rewording to 'evidence of hemishperic surface temperature asynchrony'.

Page 2 line 7: above 40S can be interpreted erroneously.

Page 2 line 15: is there a difference between non-synchronous and asynchronous or are they equivalent?

Page 2 line 16: ice core records are not 'summer', correct?

Page 2 line 22: Not all models used by Bakker et al. (2013) are of intermediate complexity.

Page 2 line 27: 'neglected to take into account', consider rewording.

Page 2 lines 28-30: not sure what the purpose is of this sentence at this place.

Page 2 line 29: mostly 'mismatch' is used instead of miss-match, consider rewording.

Page 3 line 17: at several places there is an underscore between the bracket after a reference and the next word ")$''_{,perhapsalatexissue.}$

Page 3 line 21: what is meant here with 'delay'? Would we expect the two hemispheres to show synchronous maximum warmth?

Page 4 line 27 and 31: year is missing after Capron et al.

Page 5 line 2: are the model values from single grid cells?

[Figure]

Page 5 line 7: what is the basis of the chosen grouping?

Page 6 line 20: Perhaps replace 1C by 1.5C in accordance with table 2.

Page 6 line 21: Where was this 1Sv of freshwater added, in the North Atlantic or in the Southern Ocean? Please clarify.

Page 6 line 34: Is that an average over the whole North Atlantic or only over the locations for which Capron et al. (2014) provide proxy-records?

Page 7 line 11: From Figure 1 an age of 127ka seems more appropriate.

Figure 4 line 7: Space missing between 'The' and 130ka.

Figure 8 line 4: (a) should not be bold I think.

References: Goelzer et al., 2016: doi:10.5194/cp-2015-175, 2016

---

## Referee Comment (RC2) · Anonymous Referee #2 · 11 Apr 2016

**Overview**

The recent review of high latitude climate records by Capron et al., (2014) showed that the early Eemian (ca 130 ka) was anomalously cool at northern high latitudes and anomalously warm at southern high latitudes (when compared to model simulations forced with orbital and GHG variations). In the current manuscript Stone et al conclude that this data versus model discrepancy is explained by meltwater discharge into the North Atlantic. To justify their conclusion they present global climate model simulations showing that the temperature discrepancy is reduced when 0.2Sv of freshwater is applied to the North Atlantic.

I congratulate the authors on their synthesis of observations and model results. Once some weaknesses are addressed I think the manuscript will make an important contribution to our understanding of Eemian climate.

I have two main concerns. First, I agree that the model results support the hypothesis that freshwater fluxes to the North Atlantic reduce the data model discrepancy. However, I do not think that this *proves* that freshwater fluxes are the explanation. Second, the authors invoke the 'bipolar seesaw mechanism' without defining what they really mean. It is not the authors fault that the bipolar seesaw mechanism is loosely defined in the literature, but I think it would help in this case to be more clear about the main process they have in mind. I expand on these points below. If the authors can address these and also the minor technical points that I raise further down then I am very happy to recommend publishing the paper in *Climate of the Past*.

**Major points**

- *Fresh water fluxes*: Freshwater fluxes (FW) are commonly applied to suppress deep water production and AMOC strength in climate models. In this case the size of the perturbation, 0.2 Sv, is supported by data. The 0.2 Sv FW flux is plausible, but I think the authors should be a little more cautious in their conclusion (e.g. in the abstract) that FW release is what 'accounts for' the observed temperature anomalies. Recent work on the timing of IRD layers, AMOC changes and temperature anomalies provide a good lesson on why such caution is advised. Barker et al., (2015) and Alvares Solas et al., (2013) have shown that the Heinrich Event 1 freshwater release into the North Atlantic and marginal seas comes *too late* to have caused the AMOC shutdown seen in proxies during the early part of the last deglaciation. Furthermore, recent papers have presented alternative triggers for AMOC changes, such as salt oscillator in the North Atlantic (Peltier and Vettoretti, 2014) or changes in Laurentide ice sheet height affecting wind-stress over the sub-polar gyre (Zhang et al., 2014). Climate changes at northern high latitudes due to shifts in modes of atmospheric circulation also remains a possibility (Kleppin et al., 2015), as appears to be the case in the NorESM simulation cited in the text (p5l21). All this is to say that while FW forcing reduces the data model discrepancy it does not rule out alternative mechanisms for triggering millennial-scale cooling of the NH; the authors need to acknowledge this in the revised version. Some discussion of alternative mechanisms would strengthen the paper.

- The HadCM3 simulations are run for 200 years. I doubt that this is long enough to see the final result of changes in ocean heat transport on Antarctic temperature. The recent work by the WAIS Divide Project Members (2015) shows that during MIS3 the *onset* of the bipolar seesaw signal in the WAIS ice core systematically lags Greenland transitions by ca 200 years (i.e. they report not seeing any signal for the first 200 years). The Antarctic warming in response to FW discharge in the North Atlantic appears to be arriving sooner than this in the HadCM3 simulations - which begs the question: how is the signal propagated so quickly to the southern high latitudes? I don't think the answer to this question is needed in the current manuscript, but the authors should at least acknowledge that Antarctic and Sth Ocn temperatures have probably not completed their adjustment to the change in ocean heat transport.

- p5l18: Stocker's (1998) perspective covers several possible mechanisms for out of phase climate changes in Antarctica and Greenland. The authors do not spell out which of these mechanism they are referring to. Is it the concept, mostly attributed to Crowley (1992), of a change in northward heat transport in the Atlantic? Or is it Broecker's (1998) idea of competition between NADW and AABW production? Some more discussion is needed here and some additional references.

**Technical and minor points**

1. Figure 4 and p4l18: It's counterproductive to begin the results section by compar-
   ing the 130k time slice with the Turney and Jones (2010) data. Three reasons:
   (1) TJ2010 is not the new result here so why put it first. (2) As is pointed out, the
   TJ2010 assumptions of synchronous temperature changes across the Eemian
   and of annual mean temperature estimates are flawed. (3) In any case, it It
   doesn't make sense to compare their 116-130ka slice to your 129-131ka mode
   time slice (as you say, any similarities are misleading!). I suggest to cover TJ2010
   in the introduction and perhaps later in the discussion, but remove from Figure 4
   and remove from the start of the results section.

2. In all figures the temperature anomalies that are not significant according to a t
   test need to be masked out.

3. p2 l17: ..early *onset of* warming..

4. p4 l32-p5l11: The flow of the results section is interrupted by the digression to
   talk about two methods of calculating RMSE. I would help the reader to focus on
   the results if the RMSE methods were moved to a subsection of the methods.

5. It appears that the RMSE is being calculated without including the uncertainty
   in the observations. Since observational uncertainties are provided by Capron
   et al there is no excuse not to make full use of them here. The observational
   uncertainty should be listed each time an RMSE is given for the data vs model
   comparison (or the equivalent data should be tabulated). Better still would be to
   give the data vs model RMSE in the form of a 95/

6. Figure 4 and 6: Please state in the caption how the anomalies are calculated.
   Compared to present day control HADCM3 run?

7. p4l2: Simulations are mentioned with FW varying from 0 to 1 Sv 'to determine
   the sensitivity of the model to FW forcing under the LIG climate regime (Fig 3)'.

But Fig 3 just shows where the FW was applied. Reading on I see that the results of the sensitivity study come up near the end of the Discussion. The choice to focus on the 0.2Sv forcing is an essential part of the experimental design and so should be justified early on. I would suggest to move these details on the model's AMOC sensitivity to the methods section and also to include a reference to the current Fig 9 in the methods section.

8. p5l12: You should mention here the 12Sv reduction in the AMOC.

9. p5l14: Is 3.3C still a significant discrepancy considering the observational uncertainty?

10. p5l30: I can not find where Lunt et al (2008) discuss the influence of AMOC changes on Sth Ocn SSTs and I can not find where Vellinga and Wood (2002) discuss changes in advective heat transport to Antarctica. Please expand or revise. Pedro et al., (2016), goes into some detail on how AMOC variations may affect Antarctic and Sth Ocn temperatures and should be cited here; they emphasise the importance of sea ice changes.

11. p6l6:'only modest'. Rephrase, since the upper estimate of 4.3m is equivalent to a rather immodest 70% of the 6 m estimate.

12. p6l16: Some more discussion of the results compared with Steig (2015) would be useful. For example, do Steig's results lend support to a collapse of WAIS already by 130ka?.

13. p6l24: The decision to replace WAIS with shrubs comes with no reference or argument about why shrubs are an appropriate land cover compared for example to bare ground (as in the Dry Valleys today). Please either justify this choice or revise, also consider whether Figure 8c is really necessary.

14. p7l6: Now it becomes more clear that changes in northward heat transport within the AMOC are what you propose explains the North Atlantic cooling. Hence the Crowley (1992) mechanism should be cited earlier.

**References**

Alvarez-Solas, J. et al. (2013), Iceberg discharges of the last glacial period driven by oceanic circulation changes. Proc. Natl Acad. Sci. USA 110, 16350-16354.

Barker, S.C. et al. (2015), Icebergs not the trigger for North Atlantic cold events, Nature 520(7547), 333.

Crowley, T.J. (1992), North Atlantic deep water cools the Southern Hemisphere, Paleoceanography 7, 489.

Broecker, W. (1998), Palaeocean circulation during the last deglaciation: A bipolar seesaw?, Paleoceanography 13, 119–121.

Kleppin, H., M. Jochum, B. Otto-Bliesner, C. A. Shields, and S. Yeager (2015), Stochastic atmospheric forcing as a cause of Greenland climate transitions, J. Clim., 28, 7741–7763.

Pedro, J. B., et al. (2016), The spatial extent and dynamics of the Antarctic Cold Reversal, Nat. Geosci., 9, 51–55.

Peltier, W. R., and G. Vettoretti (2014), Dansgaard-Oeschger oscillations predicted in a comprehensive model of glacial climate: A "kicked" salt oscillator in the Atlantic, Geophys. Res. Lett., 41, 7306–7313.

WAIS Divide Project Members (2015), Precise interpolar phasing of abrupt climate change during the last ice age, Nature, 520, 661–665.

Zhang, X., M. Prange, U. Merkel, and M. Schulz (2014), Instability of the Atlantic overturning circulation during Marine Isotope Stage 3, Geophys. Res. Lett., 41, 4285–4293.

---

## Author Comment (AC1) · 7 Jun 2016

Please see below a response to the reviewer comments on our paper. We thank both reviewers for their constructive comments, which have improved the paper. Please note that we will provide the revised Figure 1 (highlighted in yellow below) following the initial decision by the Editor.

**Reviewer 1:**

An important contribution and the most novel part of this work, is the simulations in which both a NH meltwater source and changes in the WAIS are simulated. Currently this topic is only introduced in the very last line of the introduction and the simulations are not even mentioned in the conclusions. I think this part of the manuscript should be discussed more comprehensively in the introduction, including previous work (e.g. Goelzer et al, 2016, Steig et al., 2015) and indications that such a WAIS collapse has occurred during or previous to the LIG. In the conclusions question could be addressed such as 'What did the WAIS simulations tell us?', or 'How should future simulations on this topic be improved in order to better address the issues discussed in the manuscript'? References: Goelzer et al., 2016: doi:10.5194/cp-2015-175, 2016

**Added to the conclusions: "Conversely, removing the WAIS in the simulations does not improve the model-data comparison in East Antarctica or the Southern Ocean. However, the lack of data coverage does not allow us to draw conclusions regarding the configuration of the LIG WAIS. ". Added to the Abstract: ". Further simulations in which the West Antarctic ice sheet is also removed lead to warming in East Antarctica and the Southern Ocean but do not appreciably improve the model-data comparison". Note that the WAIS simulations are not the main focus of the paper, but we do think it is important to report them, to guide future workers in this field, even though we can draw no strong conclusions from them. Goelzer et al is currently under review rather than published, therefore we prefer not to cite that paper. In introduction: "We further perform an idealized simulation with the WAIS removed to test whether this has any additional influence on regional warming in our model framework, as recent work has indicated that some of the warmth seen in Antarctic ice core records during the LIG could partly be explained by a reduced West Antarctic ice sheet (Steig et al, 2015)". Also in results: "Indeed, a recent study has suggested that the water isotopic data from the Mount Moulton ice core drilled in West Antarctica compared with water isotopic profiles from East Antarctic ice cores, is consistent with a collapse of the WAIS during the LIG (Steig et al., 2015). This potential melting of the WAIS during the early LIG could explain or partially explain the mismatch between the model simulations and Southern Ocean/East Antarctic data timeslices at 130 ka"**

Page 1 line 17: As discussed in the manuscript, a number of previous studies used a model-data approach to investigate the impact of Northern Hemisphere freshwater forcing on the early LIG climate, so perhaps 'for the first time' is a little too strong, although this study is certainly more thorough and presents, as is mentioned, a more 'integrated model-data approach'.

**We have modified the text to: "This integrated model-data approach, the most thorough to date, provides evidence that Northern Hemisphere freshwater forcing is an important player in the evolution of early Last Interglacial climate."**

Page 1 line 22: If the LIG is from 129-116ka, can one still consider 130ka as early LIG? Perhaps a technical detail, but on the other hand a good illustration of the broader issue that defining deglacial and interglacial periods is not trivial and perhaps not even desirable.

**Yes, 130 ka can be considered as early LIG since, as pointed out by the reviewer, it is not trivial to define interglacials. The interval "129-116 ka" we provided in the submitted manuscript at the beginning of our text is based on eustatic sea level variations using the 0 m sea level value as used in the last IPCC assessment report (Masson-Delmotte et al. 2013). However, if we were to define the LIG considering an alternative way such as the one considering the ice core dD 403‰ threshold value, it would give a date of about ~132 ka (based on the AICC2012 chronology, Bazin et al. 2012). In their recent paper, Govin et al. (2015) clearly illustrate how the timing of the beginning of the LIG**

varies widely depending on the climatic archives and tracers that are considered. A thorough discussion on this topic is also provided in the Past Interglacials Working Group of PAGES, 2016). Still, we think it is necessary to give at the start of the paper an indication of the age interval for the LIG, and as such write in the revised manuscript: "Peak high latitude temperatures were several degrees warmer during the Last Interglacial (LIG, approximately 129-116 thousand years ago, ka, based on eustatic sea level variations, Masson-Delmotte et al. 2013) (Clark and Huybers, 2009; Masson-Delmotte et al., 2011; Otto-Bliesner et al., 2006; Sime et al., 2009; Turney and Jones, 2010)." We also add at the beginning of Section 2: "The LIG starts at 129 ka when using a definition based on the eustatic sea level (Masson-Delmotte et al. 2013); however, considering dating uncertainties associated with paleoclimatic records during this time interval (see Govin et al. 2015 for a review), and the fact that defining the boundaries of interglacial periods is not trivial (see discussion in the PIGS Working Group of PAGES, 2016), we consider our 130 ka simulations as representative of the "early LIG".

Page 2 line 14: What is meant with 'partially account for changes in seasonality of precipitation'? Do the uncertainty estimates account for this or can part of the ice core oxygen isotope changes be accounted for by changes in the seasonality of precipitation?

We apologize if the sentence in the original manuscript was unclear. Precipitation intermittency, changes in moisture origin as well as site elevation and ice origin changes (e.g. Jouzel et al., 2003; Stenni et al., 2010) probably affect the quantified temperature changes reconstructed based on ice core water isotopic records. Here, the Antarctic temperature reconstructions provided in the 130 ka data based time slice and which are the ones published in Masson-Delmotte et al. (2011) are based on the present day spatial relationship between the ice isotopic composition of the snow and surface temperature ("isotopic thermometer") after correction for sea water isotopic composition and moisture source correction taking into account deuterium excess data. An uncertainty of about 1°C can be associated to these reconstructions. However, these reconstructions are considered as annual means while in principle they reflect precipitation-weighted temperatures. Bias due to possible changes in the seasonality of precipitation cannot be quantified in ice core data but in order to account at least partially for them, we consider an overall uncertainty of 1.5°C for the Antarctic temperature reconstructions included in the 130 ka data time slice. We have now slightly reorganized the paragraph and we have rephrased the sentence in the revised manuscript, we hope this is clearer now: "(see Capron et al. (2014) for methodological details and 2σ uncertainty estimates for individual records). Note that Antarctic annual surface air temperature reconstructions are estimated based on the water isotopic records after correction for sea water isotopic composition and moisture source correction using deuterium excess data (Masson-Delmotte et al. 2011). Capron et al. 2014 consider an error of 1.5°C associated with these reconstructions. It accounts for the uncertainty associated with this method and also partially accounts for the uncertainty associated with possible impacts of changes in seasonality of precipitation on the reconstructions, which remains difficult to quantify in ice core data (Masson-Delmotte et al. 2011)."

Page 3 lines 7-9: It is mentioned that Loutre et al. (2014) already performed a model-data comparison including NH freshwater fluxes for the early LIG, but that their work still showed model-data mismatches. Doesn't the present work still show these? Perhaps they became smaller? Or we have a better understanding of why these mismatches occur?

We agree that the present work does show the mismatches highlighted in Loutre et al. (2014). It is difficult to determine if these mismatches are smaller overall or not as Loutre et al (2014) only compared with a set of surface temperature timeseries from only 12 locations while we are looking at a time slice averaged over 2ka and centred on 130 ka .

Page 3 lines 16-19: Both Bakker et al. (2013) and Loutre et al. (2014) included a model-data comparison, be it small and less rigorous than the one presented here. Another such model-data comparison for this time interval was performed by SanchezGoni et al. (2012).

**Bakker et al. (2013) only included a model inter-comparison. No comparisons were made with data. Loutre et al. (2014) do not shown a comparison with the data sets in the Southern Hemisphere given in Capron et al. (2014). Sanchez-Goni et al (2012) only compare with one record from the North Atlantic. We have, however, included the Sanchez-Goni (2012) record and modified the text for clarity: "Although previous modeling studies (e.g. Bakker et al., 2013; Holden et al., 2010; Loutre et al., 2014; *Sanchez-Goni et al., 2012*) have looked at the impact of freshwater forcing on early LIG climate they did not link the response with the data reconstructions in the high latitude *regions of the Northern and Southern Hemispheres…*"**

Introduction: The recent work by Goelzer et al. (2016) should be discussed as well since it is closely related to the questions that are addressed in this manuscript.

**The Goelzer paper is currently under review rather than published, therefore we prefer not to cite that paper.**

Experimental design: Discuss some aspects of the experimental design in a little more detail: Are 200 year simulations are sufficiently long to investigate a bi-polar seesaw response? Hosing a large region between 50-70N seems highly idealized. What do we know about the distribution of meltwater during that period and what difference could it make to include a more realistic meltwater scheme? Meltwater from the WAIS is neglected. Why and how could this impact the results?

**Added "To test the robustness of the results to the 200-yer simulation length, we extended the 130 ka simulation with 0.2 Sv of freshwater forcing for a further 200 model years (400 years in total). In the Southern Ocean the rate of change of summer-SST with time is very small, and the difference between the 50-yr climate mean JFM anomaly after 200 years compared with the 50-yr climate mean after 400 years is trivial (not shown); the difference ranges between -0.5 and 0.5°C for the majority of the region, which is well within the uncertainty of the data synthesis from Capron et al. (2014) of 2.6°C on average". Added "Given the uncertainty around the actual location of the freshwater flux, we prescribe an idealized hosing region". Added "Given the uncertainty in the location and rate of freshwater forcing associated with the WAIS removal, we do not prescribe additional freshwater fluxes from the WAIS. "**

Page 4 lines 28-30: What about model uncertainties or inter-model differences in simulated temperature anomalies, can those explain the model-data mismatch?

**The point we are making here is that in our model, the model-data discrepancy is much too large to be explained by uncertainties associated with the surface temperature reconstructions from the marine and ice records (2σ of 2.6°C on average). Later in the paper we discuss the inter-model differences.**

Page 6 lines 4-16: It does not really become apparent from this paragraph that another important reason to perform simulation in which the WAIS is removed is because this could explain the persisting SH model-data mismatch.

**We agree with the reviewer and have added the following sentence: "This potential melting of the WAIS during the early LIG could explain or partially explain the mismatch between the model simulations and Southern Ocean/East Antarctic data timeslices at 130 ka."**

Page 6 line 18: From table 2 it appears to me that the number are identical so why is the model-data match slightly improved?

**The reviewer is correct that the numbers are identical to 1 d.p. As a result we have reworded the text.**

Page 6 line 24: Why would you replace it with shrubs? Is there any indications that those would grow there during the LIG? And related to that, why is such a big impact found between replacing it with bare ground or shrubs, I would expect that that region is covered with snow year round?

**Added "There is some uncertainty as to the extent or type of vegetation which may or may not have grown on an unglaciated West Antarctica during the LIG, and the vegetation type replacing a previously glaciated surface can have significant effect on the magnitude of warming (Stone and Lunt, 2013).". Also in Methods, added ", to test the response to uncertainty in the land-cover type which would replace the ice sheet"**

From the last paragraph of the results section and figures 4 and 6 it is not fully clear to me how SH temperatures evolved during the LIG and how this relates to the limited NH freshwater forcing after 127ka. This scenario would suggest that after the early LIG, when the NH freshwater returned to a low baseline, the bi-polar seesaw seized, potentially leading to cooling in the SH. Is that seen in the 125ka time-slice of Capron et al. (2014)? If not, how could this be explained? Please discuss this very interesting topic in more detail in this paragraph and perhaps include suggestions for future research on this topic.

**The Capron et al (2014) data shows that at 125 ka, the temperatures around Antarctica are still relatively warm, whereas the North Atlantic is no longer cold, relative to 130 ka. This is not very surprising when looking at Figure 4, because the freshwater has a much larger cooling effect in the Northern Hemisphere than it has warming effect in the Southern Hemisphere. Added "However, in order to fully explore the temporal variations in temperature through the LIG, fully transient simulations with time-evolving forcings would be required.**

Table 2: The improvement of the SH and EAIS model-data match when included the NH 0.2Sv meltwater forcing is surprisingly small. What are we missing?

**This is simply related to the fact that the freshwater has a large cooling effect in the north Atlantic, but a relatively minor warming over Antarctica itself and the Southern Ocean.**

Table 2: The lowest two lines (125ka) are they also compared with the 125ka time-slice of Capron et al. (2014)? Please explain in the caption.

**Yes the 125 ka experiments are also compared with Capron et al. (2014). We have added the following text to the figure caption: "The model output is compared with the 130 ka and 125 ka time slices from Capron et al. (2014)."**

Table 2: Mention in the caption which simulations were previously published and which are newly performed for this study.

**We have added the following text for clarity: "Note that the simulations without freshwater forcing includeed were previous described in Lunt et al. (2013) and references therein."**

Figure 1: Consider including a proxy records showing the AMOC evolution during this period. For instance one of the d13C records shown by Sanchez-Goni et al. (2012) and Govin et al. (2012).

**We will add in the revised Figure 1 the δ$^{13}$C record from the North Atlantic core CH69-K09 from Govin et al. (2012)**

Figure 1: An additional vertical axis showing the rate of sea level change in Sv would be easier to compare with for instance figure 9.

**We will add this to the revised Figure 1.**

Figure 1: the 'early SH warmth' is not very clear in EDC temperatures. Please clarify.

**We will enlarge the vertical axis of the EDC temperature reconstruction to make the early Antarctic Warming more obvious in the revised Figure. We also now state in the caption of the revised**

manuscript the following: "Note that Govin et al. 2015 reports in the Table 5 of their paper that the Antarctic reconstructed surface temperature (based on EDC dD) starts increasing at 135.6 ±2.5 ka based on the use of the RAMPFIT software."

Figure 1: please include in the caption a description of the grey band shown in the figure.

**Added to caption: The grey band highlights the 129-131 ka time interval that has been considered for the construction of the 130 ka data based time slice for surface temperature (see Capron et al. 2014 for details on the methodology).**

Figure 3: This could perhaps also be shown in another figure that shows a map of the North Atlantic region.

**We think that the region is most clearly represented in this way – adding it to the other Figures would make them somewhat cluttered.**

Figure 6: Why is the model response so different over Antarctica while it is so similar over the Southern Ocean? Is seems unrelated to the changes in the North Atlantic. Is this difference also there at 130ka?

**Note that left and right panels are two different models in Figure 6. They have different oceans and different land surface and seaice schemes. As such, it is not particularly surprising that they exhibit different responses over the ocean compared to over land.**

Figure 8: Why does the North Atlantic show a warming in figure c?

**This warming is relatively weak – as such we do not think it is a particularly robust signal, and likely to be model-dependent. We consider it beyond the scope of this paper to explore this small signal in detail.**

Page 1 lines 13-17. Line is very long and difficult to read. Please rewrite.

**We have re-worded this section accordingly: "Using a full complexity General Circulation Model we perform climate model simulations representative of 130 ka conditions which include a magnitude of freshwater forcing derived from data at this time. We show that this meltwater from the remnant Northern Hemisphere ice-sheets during the glacial-interglacial transition accounts for the observed colder than present temperatures in the North Atlantic at 130 ka and also results in warmer than present temperatures in the Southern Ocean via the bipolar seesaw mechanism."**

Page 1 line 22 and 27: At multiple locations double brackets are used, either like (...(..)) or like (...)(..). Consider adjusting.

**We have edited this to avoid )( and (…(..))**

Page 2 line 2: consider removing 'build'.

**We have changed this to: "However, such a unique time slice representative of LIG maximum warmth…"**

Page 2 line 5: consider rewording to 'evidence of hemishperic surface temperature asynchrony'.

**Done.**

Page 2 line 7: above 40S can be interpreted erroneously.

**We agree and have reworded accordingly: "(latitudes northward of 40°N and southward of 40°S)"**

Page 2 line 15: is there a difference between non-synchronous and asynchronous or are they equivalent?

**These are equivalent but we have changed this to asynchronous for consistency.**

Page 2 line 16: ice core records are not 'summer', correct?

**The reviewer is correct and this is not clear in the text. We have now added annual in brackets after Antarctic.**

Page 2 line 22: Not all models used by Bakker et al. (2013) are of intermediate complexity.

**We have reworded this sentence to reflect this: "An ensemble of LIG transient simulations with climate models of intermediate complexity or GCMs with low resolution/accelerated forcing,…"**

Page 2 line 27: 'neglected to take into account', consider rewording.

**We have changed the text to: "For example, previous GCM simulations did not consider freshwater forcing…"**

Page 2 lines 28-30: not sure what the purpose is of this sentence at this place.

**This sentence is included to illustrate that this missing process of freshwater forcing from melting ice sheets has also been shown to account for a mismatch between data and model to back-up why this should be explored for the early LIG. We have inserted the word "Accordingly" at the beginning of the sentence to improve the linkage.**

Page 2 line 29: mostly 'mismatch' is used instead of miss-match, consider rewording.

**We have changed this to mismatch.**

Page 3 line 17: at several places there is an underscore between the bracket after a reference and the next word ")00,perhapsalatexissue.

**This has now been rectified.**

Page 3 line 21: what is meant here with 'delay'? Would we expect the two hemispheres to show synchronous maximum warmth?

**We have changed this to "difference" in peak warmth rather than delay. The astronomical forcing at 130ka would suggest that you would expect to see warming in the Northern Hemisphere earlier than shown in the data.**

Page 4 line 27 and 31: year is missing after Capron et al.

**The years have been inserted.**

Page 5 line 2: are the model values from single grid cells?

**Yes the model values are taken from a single grid-cell. We have clarified this in the text.**

Page 5 line 7: what is the basis of the chosen grouping?

**Added "(chosen based on geographical proximity)".**

Page 6 line 20: Perhaps replace 1C by 1.5C in accordance with table 2.

**We prefer to keep 1˚C as the value in Table 2 refers to the RMSE.**

Page 6 line 21: Where was this 1Sv of freshwater added, in the North Atlantic or in the Southern Ocean? Please clarify.

**We have clarified this by stating in the "North Atlantic"**

Page 6 line 34: Is that an average over the whole North Atlantic or only over the locations for which Capron et al. (2014) provide proxy-records?

**Yes, the average is only over the locations for which Capron et al. (2014) provide records in the North Atlantic. We have modified the text accordingly: "Figure 9 shows the *model* summer North Atlantic temperature response *(averaged over the locations for which Capron et al. (2014) provide temperature records)* for freshwater input varying from 0 to 1 Sv compared with the average NH temperature anomaly from the Capron et al. (2014) dataset (horizontal dashed line)"**

Page 7 line 11: From Figure 1 an age of 127ka seems more appropriate.

**We prefer to keep 128 ka, which seems reasonable from Figure 1.**

Figure 4 line 7: Space missing between 'The' and 130ka.

**Done.**

Figure 8 line 4: (a) should not be bold I think.

**Done.**

**Reviewer 2**

Fresh water fluxes: Freshwater fluxes (FW) are commonly applied to suppress deep water production and AMOC strength in climate models. In this case the size of the perturbation, 0.2 Sv, is supported by data. The 0.2 Sv FW flux is plausible, but I think the authors should be a little more cautious in their conclusion (e.g. in the abstract) that FW release is what 'accounts for' the observed temperature anomalies. Recent work on the timing of IRD layers, AMOC changes and temperature anomalies provide a good lesson on why such caution is advised. Barker et al., (2015) and Alvares Solas et al., (2013) have shown that the Heinrich Event 1 freshwater release into the North Atlantic and marginal seas comes *too late* to have caused the AMOC shutdown seen in proxies during the early part of the last deglaciation. Furthermore, recent papers have presented alternative triggers for AMOC changes, such as salt oscillator in the North Atlantic (Peltier and Vettoretti, 2014) or changes in Laurentide ice sheet height affecting windstress over the sub-polar gyre (Zhang et al., 2014). Climate changes at northern high latitudes due to shifts in modes of atmospheric circulation also remains a possibility (Kleppin et al., 2015), as appears to be the case in the NorESM simu- lation cited in the text (p5l21). All this is to say that while FW forcing reduces the data model discrepancy it does not rule out alternative mechanisms for triggering millennial-scale cooling of the NH; the authors need to acknowledge this in the revised version. Some discussion of alternative mechanisms would strengthen the paper.

**In the Abstract, changed "accounts for" to "produces a modelled climate response similar to".**

The HadCM3 simulations are run for 200 years. I doubt that this is long enough to see the final result of changes in ocean heat transport on Antarctic temperature. The recent work by the WAIS Divide Project Members (2015) shows that during MIS3 the *onset* of the bipolar seesaw signal in the WAIS ice core systematically lags Greenland transitions by ca 200 years (i.e. they report not seeing any signal for the

first 200 years). The Antarctic warming in response to FW discharge in the North Atlantic appears to be arriving sooner than this in the HadCM3 simulations - which begs the question: how is the signal propagated so quickly to the southern high latitudes? I don't think the answer to this question is needed in the current manuscript, but the authors should at least acknowledge that Antarctic and Sth Ocn temperatures have probably not completed their adjustment to the change in ocean heat transport.

**See response to similar comment from Reviewer 1: Added "To test the robustness of the results to the 200-year simulation length, we extended the 130 ka simulation with 0.2 Sv of freshwater forcing for a further 200 model years (400 years in total). In the Southern Ocean the rate of change of summer-SST with time is very small, and the difference between the 50-yr climate mean JFM anomaly after 200 years compared with the 50-yr climate mean after 400 years is trivial (not shown); the difference ranges between -0.5 and 0.5˚C for the majority of the region, which is well within the uncertainty of the data synthesis from Capron et al. (2014) of 2.6˚C on average.**

p5l18: Stocker's (1998) perspective covers several possible mechanisms for out of phase climate changes in Antarctica and Greenland. The authors do not spell out which of these mechanism they are referring to. Is it the concept, mostly attributed to Crowley (1992), of a change in northward heat transport in the Atlantic? Or is it Broecker's (1998) idea of competition between NADW and AABW production? Some more discussion is needed here and some additional references.

**Yes, correct – the mechanism is most similar to that proposed by Crowley. Changed to "The addition of freshwater into the North Atlantic results in a bipolar seesaw response (Stocker, 1998) with a redistribution of heat between the hemispheres resulting from decreased northward heat transport through the Atlantic (Crowley, 1992)"**

Figure 4 and p4l18: It's counterproductive to begin the results section by comparing the 130k time slice with the Turney and Jones (2010) data. Three reasons: (1) TJ2010 is not the new result here so why put it first. (2) As is pointed out, the TJ2010 assumptions of synchronous temperature changes across the Eemian and of annual mean temperature estimates are flawed. (3) In any case, it It doesn't make sense to compare their 116-130ka slice to your 129-131ka mode time slice (as you say, any similarities are misleading!). I suggest to cover TJ2010 in the introduction and perhaps later in the discussion, but remove from Figure 4 and remove from the start of the results section.

**We think that it is important to show our results in the context of previous work. Figure 4 is specifically designed to show the transition (from left to right) of (a to b) improving the interpretation of the data and the seasonality of the models, and (b to c) adding freshwater to the models. We think that this transition is best represented by showing the Turney et al data, even if we do argue that its interpretation is flawed.**

In all figures the temperature anomalies that are not significant according to a t test need to be masked out.

**In this case we only discuss anomalies in the text which are substantial, and therefore not likely to be an artefact of the interannual variability. In addition, showing the entire signal rather than masking out can be informative as to the spatial structure and extent of the anomlies. Furthermore, the t-test is not appropriate unless the underlying data is normally distributed, which it rarely is in terms of climatic data. As such, we prefer not to mask out regions as suggested.**

p2 l17: ..early *onset of* warming..
**Done.**

p4 l32-p5l11: The flow of the results section is interrupted by the digression to talk about two methods of calculating RMSE. I would help the reader to focus on the results if the RMSE methods were moved to a subsection of the methods.
**Done.**

It appears that the RMSE is being calculated without including the uncertainty in the observations. Since observational uncertainties are provided by Capron et al there is no excuse not to make full use of them here. The observational uncertainty should be listed each time an RMSE is given for the data vs model comparison (or the equivalent data should be tabulated). Better still would be to give the data vs model RMSE in the form of a 95/

**Added "Note that the Capron et al (2014) dataset cites uncertainties in the data of 2.6ºC on average for the data, and the RMSE values should be viewed in this context." RMS statistics do not account for the uncertainty in the observations. To do so would probably require some sort of Bayesian calculation which is beyond the scope of this work.**

Figure 4 and 6: Please state in the caption how the anomalies are calculated. Compared to present day control HADCM3 run?

**Added "Anomalies calculated relative to the preindustrial for the model and relative to modern for the data."**

p4l2: Simulations are mentioned with FW varying from 0 to 1 Sv 'to determine the sensitivity of the model to FW forcing under the LIG climate regime (Fig 3)'. But Fig 3 just shows where the FW was applied. Reading on I see that the results of the sensitivity study come up near the end of the Discussion. The choice to focus on the 0.2Sv forcing is an essential part of the experimental design and so should be justified early on. I would suggest to move these details on the model's AMOC sensitivity to the methods section and also to include a reference to the current Fig 9 in the methods section.

**Moved the reference to Figure 3 earlier to clarify that it relates only to the location of the freshwater flux. Referenced Figure 9 in the Methods. We justify the reason for focussing on 0.2 Sv in the methods: "According to the highly-resolved millennial-scale global sea level reconstruction based on Red Sea records (Grant et al., 2012) the rate of sea level rise was 21.8 m/kyr at 130 ka during the glacial-interglacial transition (Fig. 1f, g). This is equivalent to a flux of approximately 0.2 Sv, an estimate in agreement with the 0.19 Sv calculated by Carlson (2008) based on coral records. As a consequence we choose a NH freshwater input (assuming no contribution from the melting of the Antarctic ice-sheet at this time) of 0.2 Sv (HadCM3_BRIS_130ka_0.2Sv) as our best-estimate scenario with which to compare our model temperature output and the high latitude data synthesis at 130 ka."**

p5l12: You should mention here the 12Sv reduction in the AMOC.
**Done.**

p5l14: Is 3.3C still a significant discrepancy considering the observational uncertainty?
**See similar comment from Reviewer 1. Added to Methods: "Note that the Capron et al (2014) dataset cites uncertainties in the data of 2.6°C on average for the data, and the RMSE values should be viewed in this context."**

p5l30: I can not find where Lunt et al (2008) discuss the influence of AMOC changes on Sth Ocn SSTs and I can not find where Vellinga and Wood (2002) discuss changes in advective heat transport to Antarctica. Please expand or revise. Pedro et al., (2016), goes into some detail on how AMOC variations may affect Antarctic and Sth Ocn temperatures and should be cited here; they emphasise the importance of sea ice changes.

**Agreed – replaced with "Recent work has suggested that the climatic signals arising from changes in the northward heat transport in the Atlantic, such as we have here, can be communicated to Antarctica by feedbacks associated with sea ice (Pedro et al, 2016). "**

p6l6:'only modest'. Rephrase, since the upper estimate of 4.3m is equivalent to a rather immodest 70% of the 6 m estimate.
**Rephrased to: "The contribution of the Greenland ice-sheet to global LIG sea level rise has recently been quantified (Born and Nisancioglu, 2012; Colville et al., 2011; Helsen et al., 2013; NEEM community members, 2013; Quiquet et al., 2013; Stone et al., 2013), with the IPCC Fifth Assessment Report stating a range very likely between 1.4 and 4.3 m of equivalent sea level height (Masson-Delmotte et al., 2013). "**

p6l16: Some more discussion of the results compared with Steig (2015) would be useful. For example, do Steig's results lend support to a collapse of WAIS already by 130ka?.
**In response to comments by Reviewer 1, we now set up the paper by expanding on the findings by Steig. However, their results cannot be used to argue strongly for the exact timing of the WAIS collapse.**

p6l24: The decision to replace WAIS with shrubs comes with no reference or argument about why shrubs are an appropriate land cover compared for example to bare ground (as in the Dry Valleys today). Please either justify this choice or revise, also consider whether Figure 8c is really necessary.

**We do think this is important. There is uncertainty as to what vegetation (if any) was present on LIG West Antarctica. Previous studies have explored this in the context of the Greenland ice sheet (e.g. Stone and Lunt, 2013). We make it clearer that this is a sensitivity study: "There is some uncertainty as to the extent or type of vegetation which may or may not have grown on an unglaciated West Antarctica during the LIG, and the vegetation type replacing a previously glaciated surface can have significant effect on the magnitude of warming (Stone and Lunt, 2013)". Also in Methods, added ", to test the response to uncertainty in the land-cover type which would replace the ice sheet"**

p7l6: Now it becomes more clear that changes in northward heat transport within the AMOC are what you propose explains the North Atlantic cooling. Hence the Crowley (1992) mechanism should be cited earlier.

**Agreed – see reply to previous comment, we now cite Crowley (1992) as suggested.**

**References**

[revised manuscript text omitted]

---

## Author Response (AR2)

**In response to the Editor comments, we swapped Figure 4 and Figure 5, and removed the "xxx" in Figure 6.**